# Strongly interacting matter exhibits deconfined behavior in massive neutron stars

Eemeli Annala [1], Tyler Gorda [2,3] ✉, Joonas Hirvonen [1] ✉,
Oleg Komoltsev [4] ✉, Aleksi Kurkela [4] ✉, Joonas Nättilä [5,6] ✉ &
Aleksi Vuorinen [1] ✉

Neutron-star cores contain matter at the highest densities in our Universe. This highly compressed matter may undergo a phase transition where nuclear matter melts into deconfined quark matter, liberating its constituent quarks and gluons. Quark matter exhibits an approximate conformal symmetry, predicting a specific form for its equation of state (EoS), but it is currently unknown whether the transition takes place inside at least some physical neutron stars. Here, we quantify this likelihood by combining information from astrophysical observations and theoretical calculations. Using Bayesian inference, we demonstrate that in the cores of maximally massive stars, the EoS is consistent with quark matter. We do this by establishing approximate conformal symmetry restoration with high credence at the highest densities probed and demonstrating that the number of active degrees of freedom is consistent with deconfined matter. The remaining likelihood is observed to correspond to EoSs exhibiting phase-transition-like behavior, treated as arbitrarily rapid crossovers in our framework.

For macroscopic systems in thermal equilibrium, the equation of state (EoS) is a central quantity that reflects not only the basic thermodynamic properties of the medium but also its active degrees of freedom and thus the underlying phase structure. For neutron stars (NSs)—extreme astrophysical objects containing the densest matter found in the present-day Universe—the EoS is closely related to their measurable macroscopic properties due to a link provided by General Relativity[1,2]. This has made NSs a unique laboratory for ultradense strongly interacting matter, with astrophysical observations informing model-agnostic studies of the EoS and attempts to determine the phase of matter inside the cores of NSs of different masses[3–24].

Quantum Chromodynamics (QCD) predicts that at very high densities, strongly interacting matter no longer resides in a phase where individual nucleons can be identified[25–28]. Instead, the active degrees of freedom become a set of elementary particles, quarks and gluons, that form a new phase dubbed quark matter (QM). In recent years, ab-initio calculations in nuclear[29–33] and particle[34–37] theory have established that the respective EoSs of nuclear matter (NM) and QM are qualitatively distinct. While the hadronic EoS is controlled and characterized by the nucleon mass scale, its QM counterpart exhibits near-conformal properties, independent of any characteristic mass scales[13,38].

[1]Department of Physics and Helsinki Institute of Physics, University of Helsinki, P.O. Box 64, FI-00014 University of Helsinki, Finland. [2]Technische Universität Darmstadt, Department of Physics, 64289 Darmstadt, Germany. [3]ExtreMe Matter Institute EMMI, GSI Helmholtzzentrum für Schwerionenforschung GmbH, 64291 Darmstadt, Germany. [4]Faculty of Science and Technology, University of Stavanger, 4036 Stavanger, Norway. [5]Center for Computational Astrophysics, Flatiron Institute, 162 Fifth Avenue, New York, NY 10010, USA. [6]Physics Department and Columbia Astrophysics Laboratory, Columbia University, 538 West 120th Street, New York, NY 10027, USA. ✉e-mail: gorda@itp.uni-frankfurt.de; joonas.o.hirvonen@helsinki.fi; oleg.komoltsev@uis.no; aleksi.kurkela@uis.no; jnattila@flatironinstitute.org; aleksi.vuorinen@helsinki.fi

This fundamental difference leaves an imprint on various thermodynamic quantities associated with the equation of state, such as the normalized trace anomaly $\Delta \equiv (\epsilon - 3p)/(3\epsilon)$, its logarithmic rate of change with respect to the energy density $\Delta' \equiv d\Delta/d\ln\epsilon$, the polytropic index $\gamma \equiv d\ln p/d\ln\epsilon$, the speed of sound squared $c_s^2 \equiv dp/d\epsilon$, and the pressure normalized by that of a system of free quarks, $p/p_{\text{free}}$. As illustrated in Table 1, these quantities indeed take markedly different values in (both low- and high-density) NM and in near-conformal QM at asymptotically high density.

At very high baryon densities, where weak-coupling methods produce reliable results, strongly interacting matter displays near-conformal properties, with conformality only mildly broken by subdominant loop effects and by the small values of the up, down, and strange quark masses. As summarized in Table 1 and discussed in detail in[13,38,39], the characteristics of this system include a small positive normalized trace anomaly $\Delta$, a similarly small but negative $\Delta'$, a sound speed just below its conformal value, $c_s^2 \lesssim 1/3$, and a polytropic index approaching its conformal limit from above, $\gamma \gtrsim 1$. Such values are common to many ultrarelativistic systems, but in stark contrast with those of the hadronic phase, also summarized in Table 1. In the latter case, be it at low densities where robust ab-initio results are available[3,29] or at higher densities where one must resort to phenomenological models (see, e.g.,[13,40]), the properties of hadronic matter are dominated by the nucleon mass scale that strongly breaks conformal symmetry. At densities exceeding approx. $3n_s$ there is no longer agreement between different model predictions, but as we discuss in detail under Methods, the vast majority of viable models remain strongly nonconformal even there.

It would clearly be very interesting to contrast the above expectations with a model-agnostic inference of the NS-matter EoS, informed by ab-initio theoretical results and astrophysical observations. With few recent exceptions[24,41,42], existing studies of this kind, however, suffer from at least one of two limitations: either they fail to take into account high-density information from perturbative-QCD (pQCD) calculations, recently demonstrated to significantly constrain the NS-matter EoS down to realistic core densities[24,43,44], or they implement observational constraints in the form of hard cutoffs, limiting the measurements that can be employed.

In this work, we remedy the above shortcomings by generalizing our earlier analyses[6,13,20] to a Bayesian framework. This enables us to take advantage of altogether 12 simultaneous NS mass-radius (MR) measurements (see the Methods section) and make quantitative statements about the likelihood of a transition from hadronic to QM within stable NSs. Our analysis is performed using two independent

frameworks, the parametric speed-of-sound interpolation of[13,20] and the non-parametric Gaussian process (GP) regression of[24,44], which utilizes the high-density constraint from pQCD in a considerably more conservative fashion. Both methods are used to construct a prior for the EoS, connecting a low-density result provided by Chiral Effective Field Theory (CEFT)[3,32] to a high-density limit given by pQCD[34,37]. This prior $P(\text{EoS})$, introduced in detail under Methods, is then conditioned using a likelihood function incorporating astrophysical measurements,

$$P(\text{EoS}|\text{data}) \quad = \quad \frac{P(\text{data}|\text{EoS})P(\text{EoS})}{P(\text{data})}, \qquad (1)$$

with $P(\text{data}|\text{EoS}) = \Pi_{i=1}^{n} P(\text{data}_i|\text{EoS})$ corresponding to the uncorrelated individual likelihoods of various NS measurements, indexed here by $i$. These data, reviewed under Methods, include mass measurements of the heaviest pulsars[45–49], the LIGO and Virgo tidal-deformability constraints from the NS-NS merger event GW170817[50,51], and a number of individual mass-radius measurements using X-ray observations of pulsating, quiescent, and accreting neutron stars by the NICER and other collaborations[17,52–57]. As described under Methods, we also vary the number of independent parameters in our parametric interpolation to verify that our results are not impacted by an overly restricted prior. Additionally, we perform one analysis with a polytropic construction to further assess the robustness of our approach. The main results from our work take the form of posterior distributions for different physical quantities evaluated at the centers of NSs of various masses, which we compare to theoretical expectations.

## Results
### Conformality criterion and theoretical expectations
The approach of an individual quantity towards its conformal value provides a necessary but not sufficient condition for the restoration of (approximate) conformal symmetry in a physical system. In order to establish the conformalization of strongly interacting matter inside NS cores, one should therefore simultaneously track several quantities and in addition ensure that the system stays conformal at higher densities. Here, a particularly useful pair turns out to be $\Delta$ and $\Delta'$. They are related to $\gamma$ and $c_s^2$ via

$$\Delta \quad = \quad \tfrac{1}{3} - \tfrac{c_s^2}{\gamma}, \quad \Delta' = c_s^2\left(\tfrac{1}{\gamma} - 1\right), \qquad (2)$$

implying that when both $|\Delta|$ and $|\Delta'|$ are small, $\gamma$ and $c_s^2$ also approach their conformal limits 1 and 1/3. In addition, a small value of $|\Delta'|$ naturally ensures that $\Delta$ will remain approximately constant at higher densities.

In order to draw a demarcation line between the non- and near-conformal regimes, we combine $\Delta$ and $\Delta'$ into a single quantity, adopting the criterion

$$d_c \quad \equiv \quad \sqrt{\Delta^2 + (\Delta')^2} < 0.2 \qquad (3)$$

for the identification of near-conformal matter at a given density. We justify the value 0.2 as follows. First, as noted in Table 1, it represents a natural choice between the values this parameter takes in NM and conformal systems. Second, at a discontinuous first-order phase transition (FOPT), where $c_s^2 = \gamma = 0$ and $\Delta' = 1/3 - \Delta$ (see Eq. (2)), the quantity $d_c$ can be shown to be bounded from below by $1/(3\sqrt{2}) \approx 0.236$, so that our criterion prevents FOPTs from masquerading as conformalized matter. We choose to round this value down to 0.2 to provide numerical tolerance for the approximate version of FOPTs (i.e. arbitrarily rapid crossovers) considered in our analysis. Finally, as shown in Table 1, this corresponds approximately to the largest value obtained for this quantity in pQCD. We note that the specific value 0.2 is a choice, but that our qualitative conclusions are insensitive to its small variations.

## Table 1 | Characteristic features of dense strongly interacting matter

|  | CEFT | Dense NM | Pert. QM | CFTs | FOPT |
|---|---|---|---|---|---|
| $c_s^2$ | ≪ 1 | [0.25, 0.6] | ≲ 1/3 | 1/3 | 0 |
| $\Delta$ | ≈ 1/3 | [0.05, 0.25] | [0, 0.15] | 0 | $1/3 - p_{\text{PT}}/\epsilon$ |
| $\Delta'$ | ≈ 0 | [−0.4, −0.1] | [−0.15, 0] | 0 | $1/3 - \Delta$ |
| $d_c$ | ≈ 1/3 | [0.25, 0.4] | ≲ 0.2 | 0 | $\geq 1/(3\sqrt{2})$ |
| $\gamma$ | ≈ 2.5 | [1.95, 3.0] | [1, 1.7] | 1 | 0 |
| $p/p_{\text{free}}$ | ≪ 1 | [0.25, 0.35] | [0.5, 1] | — | $p_{\text{PT}}/p_{\text{free}}$ |

Values of a set of six dimensionless quantities characterizing the properties of dense strongly interacting matter in five different limits: "CEFT", referring to predictions for sub-saturation-density nuclear matter[3,29]; "Dense NM" referring to typical nuclear-matter model predictions at the highest density ($n \approx 3n_s$) where most models still agree with each other, roughly corresponding to the centers of typical $1.4M_\odot$ pulsars; "Pert. QM" referring to pQCD calculations at $n \gtrsim 40n_{\text{sat}}$ with $n_{\text{sat}} \approx 0.16\text{fm}^{-3}$ the nuclear saturation density[34,37]; "CFTs" referring to conformal field theories in 3+1 dimensions; and "FOPT" referring to a system undergoing a first-order phase transition. The indicated intervals should be considered approximate, while the symbol "—" implies the absence of any constraints and $p_{\text{PT}}$ in the FOPT column refers to the (constant) value of the pressure at the phase transition.

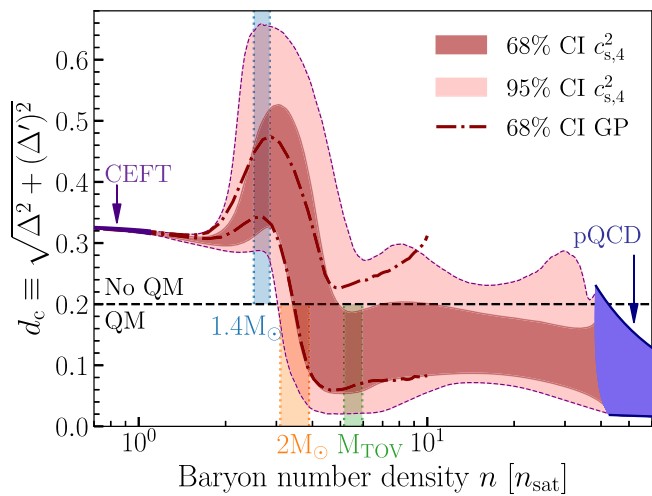

**Fig. 1 | Conformalization of neutron-star matter.** The measure of conformality $d_c \equiv \sqrt{\Delta^2 + (\Delta')^2}$, as a function of baryon density. The dark and light red bands correspond to 68% and 95% credible intervals (CIs) obtained using a 4-segment speed of sound interpolation ($c_{s,4}^2$), while the dash-dotted lines show the corresponding 68% CIs for a GP regression. Theoretical limits for $d_c$ arising from CEFT and pQCD are shown as violet bands, while the inferred quantity is seen to exhibit a qualitative change in its behavior around $n \sim 2 - 3 n_{\text{sat}}$. The blue, yellow, and green bands show 68% credible intervals for the central densities of $1.4 M_\odot$, $2.0 M_\odot$, and maximally massive $M_{\text{TOV}}$ stars, respectively. The dashed horizontal line finally corresponds to our definition of nearly conformal matter. The $1.4 M_\odot$ stars lie firmly above the dashed line, while maximally massive stars lie mostly below the line.

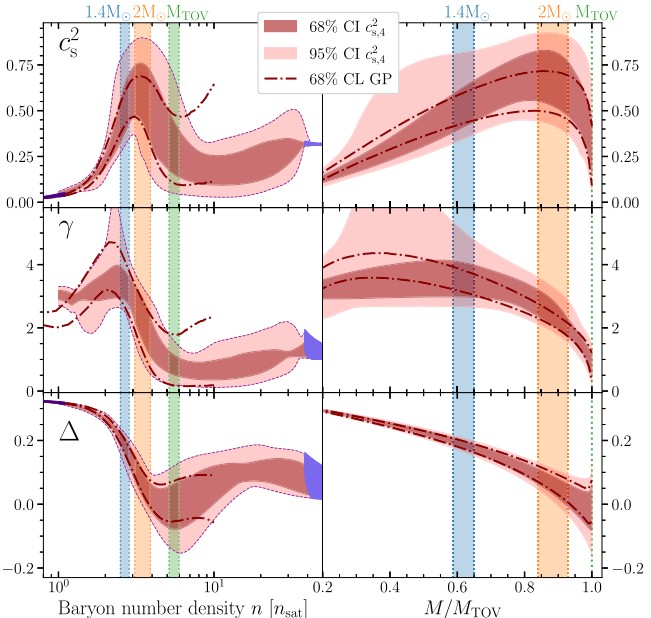

**Fig. 2 | Density dependence of neutron-star-matter properties.** The normalized trace anomaly $\Delta$, polytropic index $\gamma$, and speed of sound squared $c_s^2$ as functions of (left) the baryon number density $n$ and (right) the stellar mass $M$ normalized by $M_{\text{TOV}}$. The dark and light red bands, dash-dotted line and colored bands carry the same meaning as in Fig. 1. The $c_{s,4}^2$ and GP methods show good agreement for all quantities below the maximal density reached in stable NSs, and all three quantities display a clear change in behavior between the core densities of $1.4 M_\odot$ and $M_{\text{TOV}}$ stars.

While QM is near-conformal, not all matter that is near-conformal is QM. Therefore, establishing conformalization of matter in the cores of NSs does not, in principle, imply the presence of the deconfined phase. One quantity not fixed by conformal symmetry is the pressure normalized by the free (non-interacting) pressure $p/p_{\text{free}}$. Its value is related to the effective number of active degrees of freedom (particle species) $N_{\text{eff}}$ in both weakly and strongly coupled systems. At weak coupling, Dalton's law states that the pressure is the sum of partial pressures, which establishes the proportionality of $N_{\text{eff}} = N_f N_c p/p_{\text{free}}$. This relation can be also derived in strongly coupled conformal field theories (CFTs)[58,59] as well as in QCD at high temperatures[60,61]. Because this quantity is sensitive to the number of degrees of freedom but insensitive to the interaction strength, we expect that QM—be it weakly or strongly coupled—will have to exhibit a slowly varying $p/p_{\text{free}}$ of order one. To this end, we may use its behavior to distinguish QM from other possible near-conformal phases at NS-core densities.

From Table 1, we see that in high-density pQCD matter $p/p_{\text{free}}$ is reduced from unity to approximately 0.6 through perturbative corrections. The normalized pressure has been extensively studied also in the context of high-temperature quark-gluon plasma (QGP), where it has played a key role in establishing that deconfined matter has been successfully produced in heavy-ion collisions (see, e.g.,[60–62]). Non-perturbative lattice simulations have shown that above the pseudo-critical temperature, its value saturates to a constant around 0.8. In other high-temperature quantum field theories in the strongly-coupled limit, $p/p_{\text{free}}$ often takes fractional values smaller than one[59,63–67], and for instance in $\mathcal{N} = 4$ Super Yang-Mills theory at infinite 't Hooft coupling, $p/p_{\text{free}} = 3/4$[59]. In certain lower-dimensional CFTs, the value of this quantity is moreover related to the central charge of the theory that counts the active degrees of freedom[58,68].

## Posterior distributions
The main results of our analysis are displayed in Figs. 1, 2, 3, 4, where we show the behavior of a number of physical quantities as functions of baryon number density, baryon chemical potential, or stellar mass.

Starting from Figs. 1, 2, we show credible intervals (CIs) for $d_c$ and the triplet $c_s^2, \gamma$, and $\Delta$ as functions of the baryon number density $n$, obtained using both a four-segment speed-of-sound interpolation ($c_{s,4}^2$) and GP regression. Note that for the $c_s^2$ interpolation, we display results obtained with a four-segment interpolation which we have found to be the optimal choice, being sufficiently versatile to describe complex EoS behaviors (see Fig. 5 in "Methods" for results with a varying number of segments) and still computationally manageable.

Concretely, using the $c_{s,4}^2$ ensemble with the very conservative criterion introduced in Eq. (3), we find the posterior probability for conformalized matter being present in maximally massive stars to be approximately 88%, while the corresponding figures for $2 M_\odot$ and $1.4 M_\odot$ NSs are only 11% and 0%, respectively. The same probability for maximally massive stars obtained using the nonparametric GP interpolation is 75%, which should be considered a robust lower limit for the quantity given the very conservative way this method handles the high-density constraint. We note that these results are quantitatively stable between different computational choices (see Methods), and that they reflect the very conservative nature of the new criterion. Had we instead used the $\gamma < 1.75$ criterion introduced in[13], the posterior probability of conformalization in maximally massive stars would have been 99.8% for the $c_{s,4}^2$ and 97.8% for the GP ensemble (see also[41]). Finally, we note that the GP method allows a quantitative examination of the impact of the pQCD constraint on our results, a topic covered more extensively in[24,69]. As seen in Fig. 6 under "Methods", this impact is substantial, reflected in the likelihood of QM cores in TOV stars dropping from 75 to 50% should one not include pQCD input in the analysis.

In Fig. 3, we next show a detailed investigation of the posterior distributions of the quantities displayed in Fig. 2. The results are shown both as two-dimensional CIs for the joint distributions between pairs of quantities and as one-dimensional probability density functions, all displayed for both the $c_{s,4}^2$ and GP ensembles. Shown in gray is finally the region corresponding to our conformal criterion $d_c < 0.2$, which is

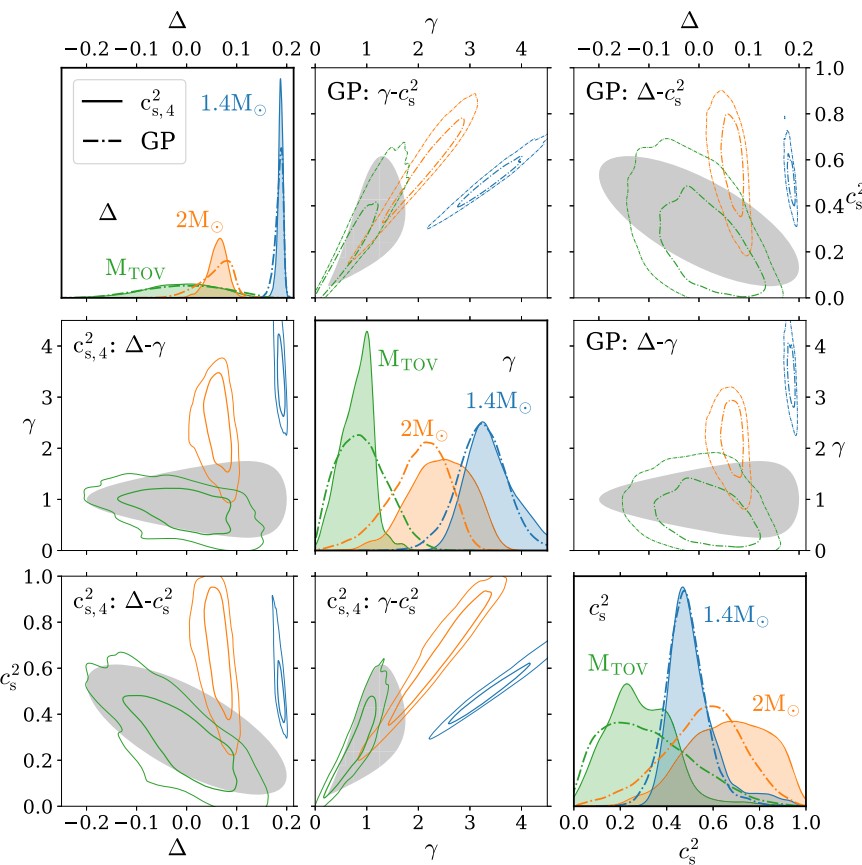

**Fig. 3 | Cross-correlation of neutron-star-matter properties.** Posterior distributions for the polytropic index $\gamma$, speed of sound squared $c_s^2$, and normalized trace anomaly $\Delta$ in the centers of NSs of different masses. The shaded gray region corresponds to our conformal criterion $d_c < 0.2$, and the two-dimensional distributions on the off-diagonal panels show 68% and 95% CIs. The one-dimensional distributions on the diagonal show the PDFs of the three quantities.

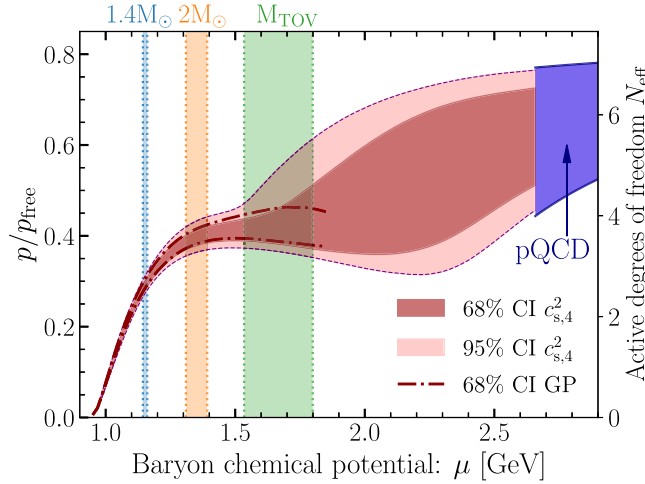

**Fig. 4 | Normalized pressure of neutron-star matter.** The pressure $p$ normalized by the free pressure $p_{free}$ as a function of baryon chemical potential $\mu$, proportional to the effective number of degrees of freedom in the system (see main text). The dark and light red bands, dash-dotted line, and colored bands carry the same meaning as in Fig. 1. This quantity is seen to exhibit a plateau within maximally massive NSs, reminiscent of an effective restoration of conformal symmetry, with a value similar in magnitude to that of weakly coupled QM.

again seen to be considerably more restrictive than the criterion $\gamma < 1.75$ used in previous works[13,41]. For $\gamma$ and $c_s^2$, we find good quantitative agreement with previous results, with the inner cores of maximally massive stars exhibiting conformalized behavior with high

credence. For all three quantities, our results as functions of $M/M_{TOV}$ are moreover in good agreement with recent works[22,38,39,42,69,70], with the approach towards conformal behavior as $M \to M_{TOV}$ clearly visible. Finally, from the $\gamma$ vs. $c_s^2$ panels, we observe that the majority of probability weight for $M_{TOV}$ stars that lies outside the gray region resides at small values of $\gamma$ and $c_s^2$. This is indicative of FOPT-like behavior, implying that should the cores of $M_{TOV}$ stars not contain conformalized matter, then they are destabilized by a FOPT, consistent with observations reported in[13].

Having established the likely conformalization of matter within the cores of maximally massive NSs, we move to an analysis of the number of active degrees of freedom. In Fig. 4, we show the behavior of the pressure normalized by its non-interacting limit, $p/p_{free}$, as a function of baryon chemical potential $\mu$. Inspecting this result, we see a clear flattening of the normalized pressure in the vicinity of the cores of maximally massive NSs, with a value $p/p_{free} = 0.40 \pm 0.03$ with 68% credence (corresponding to $N_{eff} = 3.6 \pm 0.3$) that is about 2/3 of the central pQCD value in Table 1. That the value of $p/p_{free}$ within maximally massive NSs is both so close to that of weakly coupled QM and appears to evolve very slowly gives us confidence in labeling this phase deconfined QM, although it should be noted that there exist a number of hadronic models, where $p/p_{free}$ obtains similar values close to the TOV density[40]. As we discuss in Methods, in a large majority of the latter models, the quantity is already decreasing at this density, which is at odds with the behavior we witness. It is, however, interesting to ask whether the slightly reduced value of the quantity within maximally massive NSs may signal that matter in these stellar cores features strongly-coupled characteristics.

Finally, let us briefly note two interesting issues that go somewhat beyond the scope of our current work. First, in[71] it was recently

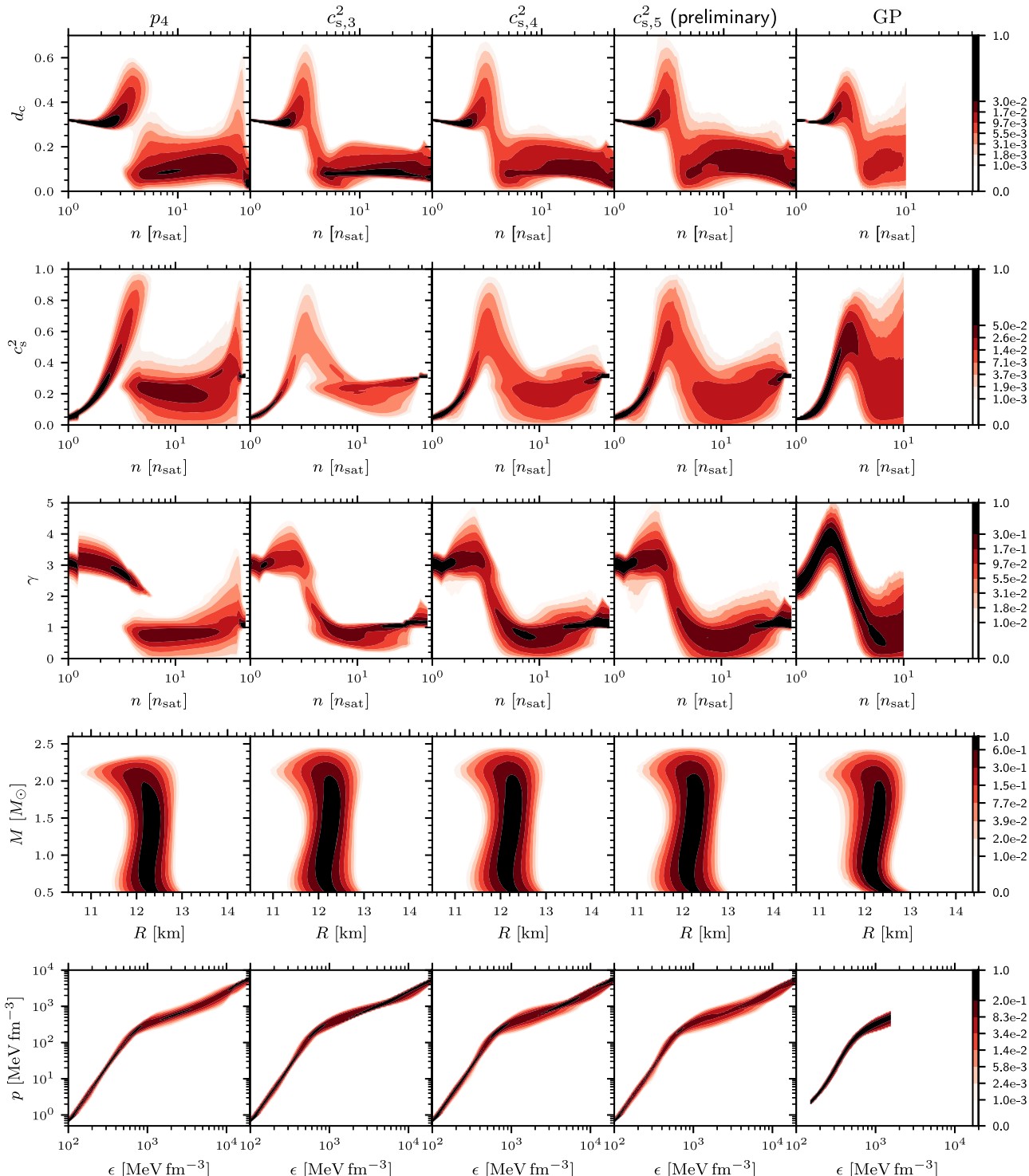

**Fig. 5 | Comparison of different interpolators.** A comparison of the parametric (interpolated) EoS model results, illustrating their robustness under varying the number of parameters as well as the form of the interpolants. Shown are always posterior CIs from the full analysis with all astrophysical constraints. The first column shows the polytropic interpolation with four segments ($p_4$). The following columns show the $c_s^2$ interpolation with an increasing number of linear segments from 3 ($c_{s,3}^2$) to 4 ($c_{s,4}^2$) to 5 ($c_{s,5}^2$), as well as the Gaussian processes parameterization (GP). For the 5-segment $c_{s,5}^2$ interpolation, the posterior sampling has not yet formally converged and, hence, the results are only indicative. The rows show the evolution of the conformal measure $d_c$, the speed of sound squared $c_s^2$, polytropic index $\gamma$, mass-radius *M-R* relation, and pressure-energy-density *p-ε*. The colorbars are tailored specifically to best highlight the variations in the probability density.

suggested that the posterior probability of QM cores sharply decreases somewhat below $M_{TOV} \approx 2.3 M_\odot$, so that if NSs more massive than this limit are discovered, QM cores can be ruled out. We have failed to reproduce this effect in our results, but instead find the probability of QM cores to have only a mild dependence of $M_{TOV}$. Second, some explicit FOPTs beginning at densities below $2n_{sat}$ have recently been shown to have a noticeable impact on hard limits for the EoS[72]. While a generalization of our calculation to include explicit discontinuities in

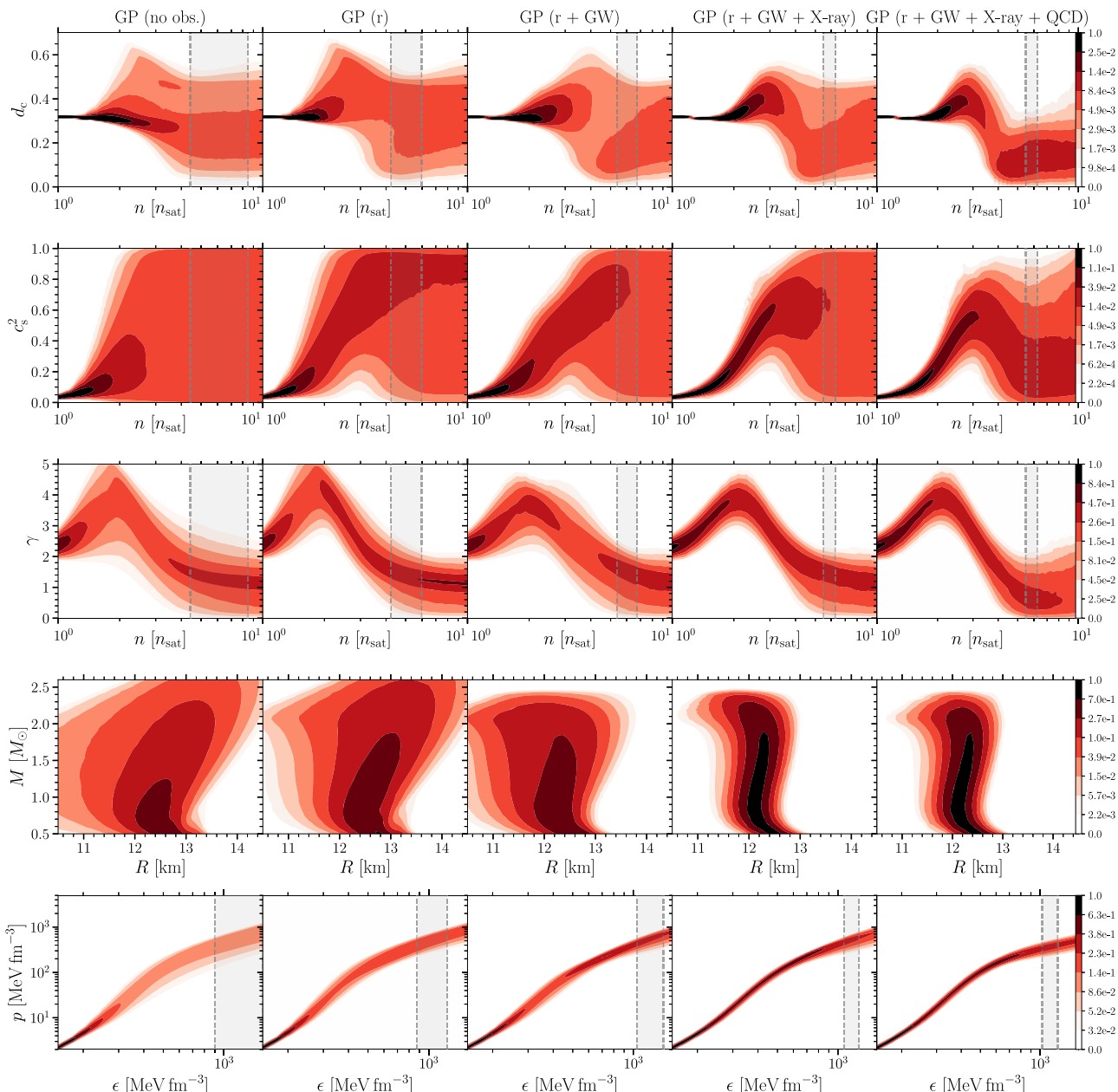

**Fig. 6 | Effect of different measurements–Gaussian processes.** Model comparison showing the effect of different measurements on the GP parameterization. The first column shows the CIs of various quantities with an EoS that does not have any astrophysical measurements nor high-density QCD input (prior). The second column shows the Bayesian posterior densities for an EoS conditioned with the $\approx 2M_\odot$ radio pulsar mass measurements. The third column shows the calculations with pulsar masses, tidal deformabilities from GW170817, as well as the assumption that the remnant in GW170817 collapsed into a black hole. The fourth column shows the calculation with pulsar masses, tidal deformabilities, the assumption that GW170817 collapsed to a BH, and X-ray measurements. The fifth column shows the calculations with all the astrophysical input from the fourth column, along with the addition of the high-density pQCD input. The gray vertical bands correspond to the density interval found at the centers of TOV-mass stars (with 68% confidence level). Other quantities and symbols as in Fig. 5.

the EoS is clearly an interesting topic for further work, we again note that the criterion for near-conformality we have adopted in this work is very conservative. Moreover, the EoSs that[72] identified to extend beyond previous hard limits were found to have very low posterior probabilities and lie right at the edge of the 90% credibility bound for the tidal-deformability constraint. We hence expect that including such EOSs with explicit FOPTs within an analysis such at our present one would result in only mild shifts in the posteriors. For this reason, we trust that the estimates we have obtained for the probability of finding conformalized matter inside massive NSs are reliable.

## Discussion

In the context of heavy-ion collisions, the properties of the equation of state (EoS) have played a key role in establishing the production of deconfined hot quark-gluon plasma. While there is no discontinuous transition in this case, the two phases can still be clearly demarcated based on their qualitatively differing material properties. At low temperatures, the EoS exhibits the qualitative features of hadronic matter while at high $T$ it is characterized by approximate conformal symmetry and a number of active degrees of freedom corresponding to deconfined quarks and gluons. Note, however, that approximate

conformal symmetry does not imply weak coupling: at the temperatures reached in collider experiments, weak-coupling methods remain poorly convergent, and the system exhibits strongly-coupled behavior. Nevertheless, the transition to QGP is identifiable from the restoration of the conformal symmetry, closely related to that of chiral symmetry.

While the high-density matter in NSs in general significantly differs from high-temperature QGP, the pattern of conformal-symmetry restoration can be similarly used as an indicator of phase change inside NSs. In this Letter, we have established that the cores of the most massive neutron stars are very likely characterized by approximate conformal symmetry. We see signs of conformalization across multiple microscopic properties characterizing the EoS, including the speed of sound $c_s^2$, polytropic index $\gamma$, normalized trace anomaly $\Delta$, and its logarithmic derivative (with respect to energy density) $\Delta'$. Moreover, we have introduced a new measure for the degree of conformality, $d_c \equiv \sqrt{\Delta^2 + (\Delta')^2}$, the behavior of which exhibits a striking transition between $1.4 M_\odot$ and maximally massive NSs, and we have determined that the number of active degrees of freedom appears to be consistent with theoretical expectations for the properties of QM. Concretely, we find that within the cores of maximally massive NSs, the ratio $p/p_{\text{free}}$ is about 2/3 of its value in weakly coupled quark matter, leading us to the conclusion that the most massive neutron stars very likely harbor cores of deconfined QM. An 88% probability (or 75% with the GP method), obtained with a very conservative definition of near-conformality, provides a quantitative estimate of the remaining uncertainties, of which the most important one is related to the possible presence of a destabilizing first-order phase transition.

To conclude, we note that in the context of heavy-ion collisions, evidence for the successful production of deconfined matter consisted of multiple arguments, of which the behavior of the EoS was but one. Similarly, it would be desirable to find other observables and arguments that may either support or contradict the evidence provided by our analysis of the EoS; for concrete suggestions, see e.g.[73–78]. Indeed, just as in the case of heavy-ion collisions, to compellingly establish the existence of QM in the cores of massive NSs, we need multiple lines of evidence all supporting the same picture.

## Methods

Our different prior EoS ensembles are constructed by joining together three different pieces: a low-density CEFT result applied up to approx. $1.2 n_{\text{sat}}$, an intermediate density interpolator, and a high-density part obtained from pQCD that is used either from $\mu = 2.6$ GeV onward or robustly translated down to $10 n_{\text{sat}}$ using the results of[43]. Here, the CEFT part depends of altogether five parameters in a way specified below, while the pQCD pressure only depends on one free parameter, the renormalization scale $\bar{\Lambda}$ of the $\overline{\text{MS}}$ scheme. Finally, in the intermediate-density region we employ two independent approaches, one built with piecewise-defined polytropes or the speed-of-sound functions of[13] and the other based on a non-parametric Gaussian process regression introduced in[24].

In the case of the parametric interpolation, we sample the parameter space with a Markov-Chain-Monte-Carlo method using the EMCEE sampler[79]. In practice, we use the affine-invariant stretch-move algorithm with 125 parallel "walkers" to generate Markov-Chain-Monte-Carlo chains with $N_{\text{chain}}/\tau \sim 3 \times 10^3$ independent samples. Here, $\tau \sim 2 \times 10^4$ is the autocorrelation time, which can sometimes be very long due to the high-dimensional parameter space. We have tested the numerical convergence of each calculation by generating sub-samples of the chains (with a jackknife re-sampling) and then verifying that the resulting credibility regions do not change. In addition, for the particular case of the four-segment $c_s^2$ interpolation we have verified the convergence by generating 50% more samples and noting that all the results remain unchanged. Finally, we smooth the posterior

distributions with kernel density estimation using Silverman's rule to estimate the kernel bandwidth.

Below, we will in turn cover the construction of our low-, high- and intermediate-density EoSs, as well as the implementation of the astrophysical constraints. After this, we perform a detailed quantitative comparison between the three frameworks used at intermediate densities (speed-of-sound and polytropic interpolation as well as Gaussian Processes), while in the last part of "Methods", we review the results of an analysis of currently available model EoSs for high-density NM.

### Low-density EoS

At low densities, up to slightly above the nuclear saturation density $n_{\text{sat}}$, the NS-matter EoS can be reliably determined using the well-tested machinery of modern nuclear theory, including the systematic effective theory framework of CEFT[29–33]. Specifically, we take the crust EoS from[80], while starting from the crust-core transition around $0.5 n_{\text{sat}}$, we use the results of[32] for pure neutron matter (PNM) based on the N3LO ab-initio calculations using the $\Lambda = 500$ MeV potentials from[31], reported up to 0.195 fm$^{-3}$, or slightly above $1.2 n_{\text{sat}}$. The upper limit for the pressure is chosen so that all (sampled) PNM EoSs have non-negative speeds of sound squared throughout the low-density interval considered, and we similarly ensure the same condition after beta equilibrium.

To transform the PNM results into a beta-stable form, we utilize the Skyrme-motivated ansatz introduced in[3], which expresses the sum of the interaction and kinetic energies in the form

$$\frac{\epsilon_1(n,x)}{T_{\text{SNM}}} = \frac{3}{5}\left[x^{5/3} + (1-x)^{5/3}\right](2\bar{n})^{2/3}$$
$$- \left[2(\alpha - 2\alpha_L)x(1-x) + \alpha_L\right]\bar{n}$$
$$+ \left[2(\eta - 2\eta_L)x(1-x) + \eta_L\right]\bar{n}^\Gamma. \quad (4)$$

Here, $x \equiv n_p/n$ denotes the proton fraction, $\bar{n} \equiv n/n_{\text{sat}}$, and $\alpha, \alpha_L, \eta$, and $\eta_L$ are parameters that can be determined from the saturation properties of symmetric nuclear matter (see[3] for discussion and numerical values). The normalization constant $T_{\text{SNM}}$ on the other hand stands for the Fermi energy of symmetric nuclear matter (SNM, $x = 1/2$) at $n = n_{\text{sat}}$,

$$T_{\text{SNM}} = \frac{1}{2m_B}\left(\frac{3\pi^2 n_{\text{sat}}}{2}\right)^{2/3} \quad (5)$$

where $m_B$ is the (mean) baryon mass, taken equal to that of the neutron $m_N \approx 939.565$ MeV. Finally, we add to Eq. (4) an additional phenomenological term,

$$\frac{\epsilon_2(n,x)}{T_{\text{SNM}}} = -\zeta_L(\bar{n} - \bar{n}_0)^4, \quad (6)$$

where we have introduced two further phenomenological parameters $\bar{n}_0$ and $\zeta_L$. Their presence can be seen to lead to a considerable improvement in the model's ability to describe the CEFT data for PNM.

Following the original study[3], we next require that the energy and pressure satisfy

1. $\epsilon(n_{\text{sat}}, 1/2) = -16$ MeV,
2. $p(n_{\text{sat}}, 1/2) = 0$, where $\epsilon = \epsilon_1 + \epsilon_2$ and $p = n^2 \partial \epsilon/\partial n$, and which aids us in the selection of the model parameters $\alpha$ and $\eta$. At the same time, we do not fix the parameter $\Gamma$ using a single value of the incompressibility parameter for SNM at saturation

$$K_s = 9 n_{\text{sat}}^2 \frac{\partial^2 \epsilon}{\partial n^2}(n_{\text{sat}}, 1/2) \quad (7)$$

as done in[3], but rather consider $\Gamma$ a free parameter following[19]. Moreover, with the mass difference between the proton and neutron

approximated to zero, we note that the proton fraction $x \equiv n_p/n$ can be solved from the equation

$$\frac{\partial \epsilon(\bar{n},x)}{\partial x} + \mu_e(\bar{n},x) = 0, \qquad (8)$$

where $\mu_e = \sqrt[3]{3\pi^2 xn}$ is the electron chemical potential in the ultra-relativistic limit[3].

In practice, we draw a large sample of the PNM EoSs of[32] using a prior distribution for the five PNM model parameters: $\alpha_L$, $\eta_L$, $\Gamma$, $\zeta_L$, and $\bar{n}_0$. This distribution is then further approximated using a Gaussian mixture model to get a simpler and smoother multidimensional distribution, which is used to represent the CEFT result in building the NS-matter EoSs.

### High-density EoS

Due to the asymptotic freedom of QCD, at very high densities the behavior of cold ($T = 0$) strongly interacting matter can be approached from the deconfined side using the machinery of modern perturbative thermal field theory. Here, we rely on state-of-the-art perturbative-QCD (pQCD) calculations for three-flavor quark matter, presented in detail in[34,37].

In the limit of beta equilibrium and strictly vanishing temperature, the pQCD pressure depends on only two parameters: the baryon chemical potential $\mu$ and the $\overline{\text{MS}}$ renormalization scale $\bar{\Lambda}$. The latter is often scaled by the central value of two times the quark chemical potential $2\mu/3$ to create a dimensionless parameter $X \equiv 3\bar{\Lambda}/(2\mu)$, which is typically varied from 1/2 to 2. For the prior probability density on $X$, we use

$$f(X) = \frac{1}{[\ln(X_{max}/X_{min})]X} \qquad (9)$$

with $X_{min} = 1/2$ and $X_{max} = 2$. The pQCD pressure is then used for baryon chemical potentials $\mu \geq \mu_{pQCD} = 2.6$ GeV, corresponding to baryon densities $n_{pQCD} \gtrsim 40 n_{sat}$.

### Intermediate densities

As noted in the main text, at intermediate densities we use two independent and complementary methods for constructing the full EoSs, largely following earlier works[13,20,24]. As none of the methods we use here is new, we leave a more extensive technical introduction to the original references.

In the case of parametric interpolation, we use both a piecewise-polytropic or piecewise-linear-$c_s^2$ EoSs with a varying number $N$ of intermediate segments between the CEFT and pQCD regimes. The resulting EoSs contain $2N + 1$ or $2N$ free parameters in the polytropic and $c_s^2$ cases, respectively.

Finally, for the nonparametric GP calculations, we exactly follow the prior construction of[24].

### Astrophysical measurements and constraints

In addition to the theoretical constraints discussed above, our EoS models have been further conditioned using astrophysical observations. We list and discuss these measurements here.

We first discuss the pulsar mass measurements. We demand that an EoS be compatible with the latest mass measurements of massive pulsars. So far, two NS systems have been observed likely containing a two-solar-mass NS, PSR J0348+0432 ($M = 2.01 \pm 0.04\, M_\odot$, 68% confidence interval)[46] and PSR J0740+6620 ($M = 2.08 \pm 0.07\, M_\odot$, 68% credible interval)[49]. We model these mass measurements as normally distributed, i.e. $M_{0432}/M_\odot \sim \mathcal{N}(2.01, 0.04^2)$ and $M_{6620}/M_\odot \sim \mathcal{N}(2.08, 0.07^2)$, respectively.

We then require that the TOV mass derived from a target EoS has to be greater than the observed masses of these two stars; otherwise, the EoS in question is discarded.

We now discuss the tidal deformability measurements. We use GW data from the first binary NS merger event GW170817 collected by the LIGO/Virgo collaboration[50]. In particular, we use the marginalized 2d posterior distribution for the tidal deformabilities of the two merger components, given in Fig. 1 of[51]. The individual tidal deformabilities for our EoSs are on the other hand calculated using formulae given in ref. [81] assuming that both compact objects are NSs. Note that the LIGO/Virgo collaboration has also detected another possible NS-NS merger event GW190425[82]. It is, however, unfortunately too faint to be of use to constrain the NS-matter EoS and is therefore omitted. Moreover, the collaboration has also detected three possible BH-NS merger events: GW190814[83], GW200105, and GW200115[84]. It is, however, very likely that GW190814 is just another BH-BH event.

In our analysis, we use two prior variables, the chirp mass of the binary

$$\mathcal{M} = \frac{(m_1 m_2)^{3/5}}{(m_1 + m_2)^{1/5}} \qquad (10)$$

and the mass ratio $q = m_2/m_1$, where $m_1 > m_2$, instead of the individual masses $m_1$ and $m_2$ of the binary components. The LIGO/Virgo collaboration has determined the binary chirp mass to be $\mathcal{M} = 1.186 \pm 0.001\, M_\odot$ (90% symmetric credibility limits)[85], which we take to be normally distributed, i.e. $\mathcal{M}/M_\odot \sim \mathcal{N}(1.186, 3.7 \times 10^{-7})$. The distribution of the mass ratio $q$ is on the other hand chosen to be uniform between 0.4 and 1, i.e. $q \sim \mathcal{U}(0.4, 1)$. Finally, we discard all parameter candidates that do not satisfy

1. $m_1$, $m_2 \leq M_{TOV}$ and
2. $\Lambda(m_1)$, $\Lambda(m_2) \leq 1600$.

We now turn to the mass-radius measurements from X-ray observations. The generated EoSs are also constrained by 12 astrophysical NS mass-radius measurements that are introduced in Table 2 together with the chosen mass priors. The corresponding radius of an individual star can be determined using the TOV equations when the EoS is generated and a candidate mass is drawn from a uniform prior distribution, $m_i \sim \mathcal{U}(m_{min}, m_{max})$, where $m_{min}$ and $m_{max}$ are the minimum and maximum masses, respectively. The limits of each mass prior are selected such that they encapsulate the complete range of the mass-radius measurement, with a maximal range of $m_i/M_\odot \in [0.5, 2.8]$. When possible, tighter limits are selected because they lead to more efficient sampling of the distribution We also ignore the additional factor to the prior, $\sim (m_{TOV} - m_{min})^{-1}$, correcting for the mass selection bias, as the factor is to a good approximation constant for our choice of $m_{min} \approx 0.5 M_\odot$[10,15].

Proceeding to the individual measurements considered, we first use the 2d mass-radius probability distributions of the pulsars PSR J0030 + 0451 and PSR J0740 + 6620 as reported by the NICER mission. The PSR J0030 + 0451 measurement is based on publicly available data from[56] employing their ST+PST model, while the PSR J0740 + 6620 measurement is similarly based on publicly available data from[17]. From the latter reference, we employ their combined NICER+XMM constraints together with the inflated cross-instrument calibration error. The differences between this choice and other possible models would result in small radius differences of $\Delta R \lesssim 0.1$ km.

In addition to the above, we also incorporate 2d mass-radius measurements from three different X-ray bursters. The most accurate constraint corresponds to the neutron star in the binary system 4U J1702 − 429 and is derived using direct atmosphere model fits to time-evolving energy spectra[54]. Here, we use their model D measurement corresponding to a fit with a free mass, radius, and composition. The other two neutron star mass-radius measurements, corresponding to

**Table 2 | Neutron-star measurements**

| System | Mass prior [$M_\odot$] | Model | Ref. |
|---|---|---|---|
| NICER pulsars | | | |
| PSR J0030+0451 | $\mathcal{U}(1.0, 2.5)$ | ST+PST | 55,56 |
| PSR J0740+6620 | $\mathcal{N}(2.08, 0.07^2)$ | N+XMM+cal. | 17,49,57 |
| qLMXB systems | | | |
| M13 | $\mathcal{U}(0.8, 2.4)$ | H | 52 |
| M28 | $\mathcal{U}(0.5, 2.8)$ | He | 53 |
| M30 | $\mathcal{U}(0.5, 2.5)$ | H | 53 |
| $\omega$ Cen | $\mathcal{U}(0.5, 2.5)$ | H | 53 |
| NGC 6304 | $\mathcal{U}(0.5, 2.7)$ | He | 53 |
| NGC 6397 | $\mathcal{U}(0.5, 2.0)$ | He | 53 |
| 47 Tuc X7 | $\mathcal{U}(0.5, 2.7)$ | H | 53 |
| X-ray bursters | | | |
| 4U 1702 − 429 | $\mathcal{U}(1.0, 2.5)$ | D | 54 |
| 4U 1724 − 307 | $\mathcal{U}(0.8, 2.5)$ | SolA001 | 86 |
| SAX J1810.8 − 260 | $\mathcal{U}(0.8, 2.5)$ | SolA001 | 86 |

A summary of the NS X-ray $M$–$R$ measurements considered in this work.

the binary systems 4U 1724 − 307 and SAX J1810.8 − 260[86], are derived using a cooling tail method analysis[87]. We do not consider other possible bursting sources because of the large uncertainty related to the unconstrained accretion disk state[88]. We also note that no rotational effects are modeled in[89]; these can introduce an additional uncertainty of order 0.5 km into the radius of a rapidly rotating source.

Lastly, we use mass-radius measurements of seven neutron stars in quiescent low-mass X-ray binary systems[52,53]. We only use systems with reliable distance measurements, which enables breaking the degeneracy between the observed flux and emitted luminosity (i.e., the size of the emitting region and the source distance). We use models that assume the surface to be uniformly emitting, indicating the absence of hot/cold spots. The atmosphere composition is selected manually (from H vs. He) using a composition that gives a radius of $R \approx 12$ km; varying the composition causes a large (~2 ×) change in $R$, so the selection of a realistic composition is relatively unambiguous for all of the used sources.

Finally, we discuss our implementation of multimessenger constraints arising from GW170817. The timing and spectral properties of the short gamma-ray burst (sGRB) and kilonova from the GW170817 event are also used to set additional constraints for the EoS. These constraints are based on the astrophysical jet generation and launching models that have demonstrated that the jet launching from a black hole merger remnant is strongly favored for sGRB events. Additionally, the prolonged existence of a supramassive/hypermassive NS (SMNS/HMNS, respectively) remnant is disfavored because of the observed moderate kinetic energy of the kilonova/sGRB: an SMNS/HMNS inside a thick disk would release substantial rotational energy into the surrounding kilonova, strongly modifying its appearance. No such modifications, such as a prolonged prompt emission phase or strong blueshifted winds, were detected from GW170817. This implies that the merger remnant either (i) collapsed immediately into a black hole or (ii) formed an SMNS or HMNS remnant that then shortly after collapsed into a black hole (see, e.g.[20] and references therein).

The short gamma-ray burst from GW170817 was detected with a lag of $\lesssim 2$ s after the NS merger, measured as the difference between the peak of the gravitational wave signal and the arrival of the first $\gamma$-rays. We therefore require that the remnant must (at least) form a supramassive NS and therefore has a total (baryon) mass exceeding the (baryon) TOV mass, $m_{b,1} + m_{b,2} > m_{b,\mathrm{TOV}}$. We assume here zero ejecta for the merger process, following previous works (see[20] for references and discussion). Additionally, we note that the short lag

of $\lesssim 2$ s favors the HMNS scenario, given that SMNSs are expected to be more long-lived with typical lifetimes of $\gtrsim 10$ s. Enforcing the HMNS scenario would lead to an even more stringent constraint, $m_1 + m_2 > m_{\mathrm{SMNS}}$, but following[20], here we only consider the more relaxed constraints following from SMNS formation.

## Comparison of the constructed EoSs

Our fiducial EoS consists of a piecewise-linear $c_s^2$ interpolant with $N = 4$ intermediate segments as a function of the baryon chemical potential $\mu$, for which we use the shorthand notation $c_{s,4}^2$. We have benchmarked the robustness of the corresponding results to other interpolation methods, including a varying number of interpolation segments, with results of this comparison displayed in Fig. 5. Comparing the parameter regions that the models can sample for $d_c, c_s^2, \gamma, p$, and the NS mass-radius relation, we find that four segments provides the minimum viable interpolation accuracy. The biggest shortcoming of the computationally most-economic three-segment model (i.e., $c_{s,3}^2$) is the lack of EoSs that mimic phase transitions at densities of $n \approx 5 - 10 n_{\mathrm{sat}}$, whereas only negligible deviations are found between the four-segment and five-segment models. We note in passing that the polytropic interpolation ($p_N$) gives very similar results when the number of segments $N$ is at least four.

A similar comparison of the effects of various astrophysical observations on our results is shown in Fig. 6 and Fig. 7, where we separately consider the GP and $c_{s,4}^2$ methods, respectively. In particular, we compare the resulting Bayesian posterior distributions for calculations with no astrophysical measurements (denoted as "no obs."); only mass-measurements from radio pulsars ("r"); pulsar mass measurements, GW deformabilities, and the SMNS hypothesis for GW170817 ("r + GW"); and finally pulsar mass measurements, GW deformabilities, the SMNS hypothesis for GW170817, and X-ray measurements ("r+GW+X-ray"). Our fiducial calculations are based on the measurement set "r+GW+X-ray", which includes all measurements. We note that especially $d_c$ does not strongly depend on the inclusion of X-ray measurements, and that there are some interesting quantitative differences between the responses of the $c_{s,4}^2$ and GP ensembles to the astrophysical measurements. This may be partially related to the fact that in Fig. 6, the high-density constraint from pQCD is only included in the fifth additional column, owing to the fact that in this approach, it is possible to turn this effect on and off at will. Once all observations and the pQCD input have been included, our final results for different physical quantities obtained using the two methods largely agree.

## Properties of nuclear matter models at high density

In this final part of the Methods section, we provide brief justification for our claim that viable models of dense nuclear matter behave differently from our results, reflected in part in the values appearing in the "Dense NM" column of Table 1. To study this question in a systematic fashion, we have analyzed a large set of publicly available nuclear-matter models, determining the characteristic ranges of various thermodynamic quantities they predict. The result of this analysis is given in Fig. 8, which display results for all hadronic $T = 0$ EoSs appearing in the CompOSE database[90]. (See also a more detailed analysis in the framework of relativistic mean-field models[91] with qualitatively similar results for $c_s^2, \gamma$ and $\Delta$.) Note that here we do not discard EoSs that fail to satisfy given hard observational limits (unlike in the discussion of[13]), but rather impose a color coding for the curves corresponding to their relative likelihoods in comparison with the most likely EoS in our GP ensemble. We also emphasize that the EoSs from the CompOSE database we analyze are not required to conform to our low-density CEFT prior; the latter appears only in the priors of our EoS ensembles and not in the likelihood function that we use here.

Our first observation from (Fig. 8a) is that the different NM models appear to be in qualitative agreement with each other for densities $n \lesssim 3 n_s$, where they all display strongly non-conformal behavior: a large

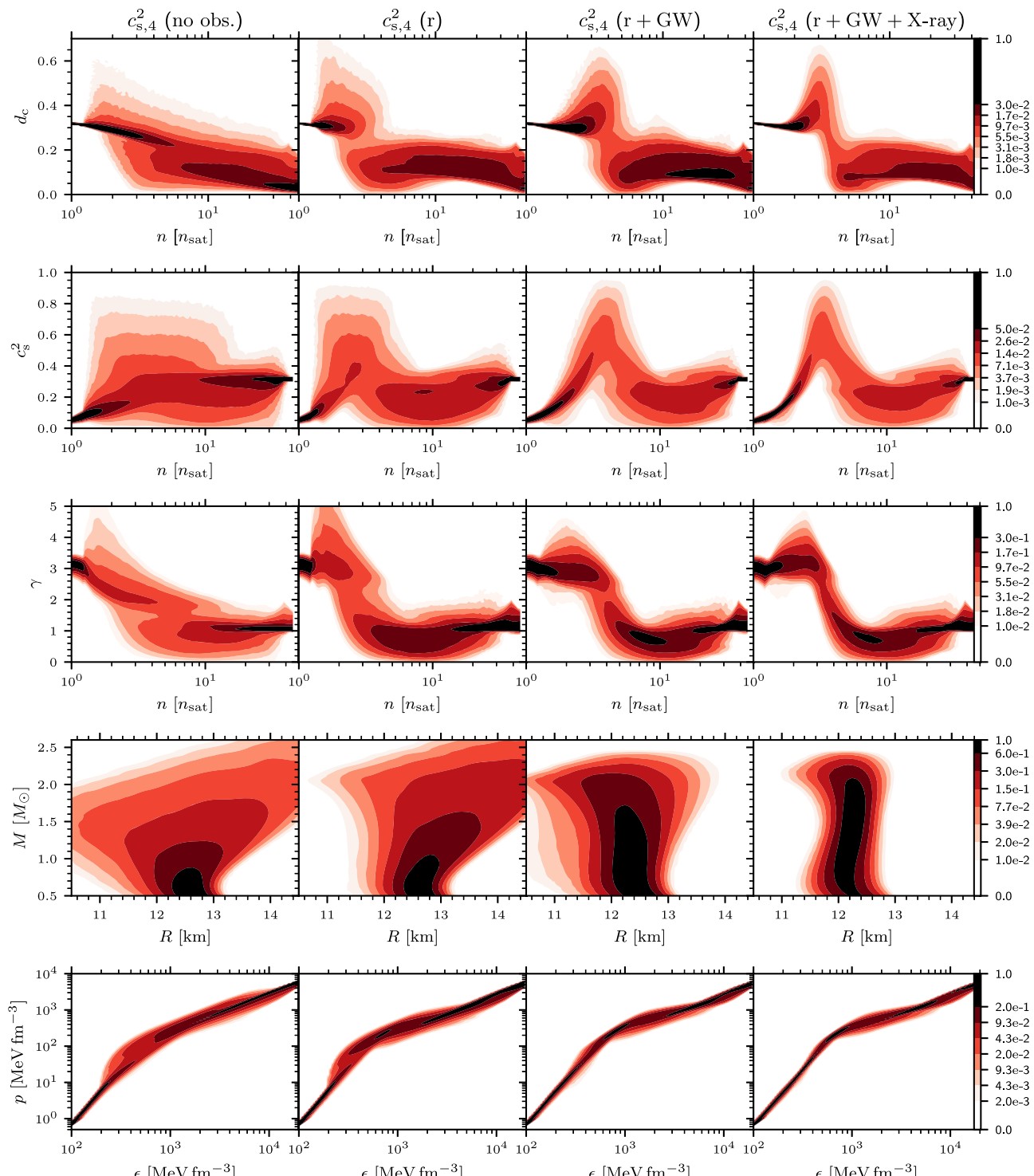

**Fig. 7 | Effect of different measurements−interpolation.** Model comparison showing the effect of different measurements on the $c_{s,4}^2$ parameterization. The first column shows the CIs of various quantities with an EoS parameterization that does not have any astrophysical measurements (prior). The second column shows the Bayesian posterior densities for EoS conditioned with the $\approx 2M_\odot$ radio pulsar mass measurements. The third column shows the calculations with pulsar masses and tidal deformabilities from GW170817, as well as the assumption that a BH was formed in GW170817. The fourth column shows the calculation with pulsar masses, tidal deformabilities, the assumption that a BH was formed in GW170817, and X-ray measurements. Other quantities and symbols are as in Fig. 5.

or rapidly changing trace anomaly, a large value of the polytropic index $\gamma$, a rapidly rising value of the speed of sound $c_s^2$, and most indicatively, a large value of the conformal distance $d_c$. Above this density, the agreement is quickly lost (visible in particular in $d_c$), which is why the ranges we report for different quantities in Table 1 are chosen to correspond to $n = 3n_s$. Proceeding towards TOV densities,

denoted by dots at the end of the curves in the four plots, some model predictions even end up at near-conformal values, but none of these models carries a non-negligible likelihood in our analysis. Discarding them, the lower end of the range we give for $d_c$ in Table 1 is observed to stay valid at all densities, so that sizable $d_c$ values can be considered a characteristic prediction of NM models. Finally, in panel b of the same

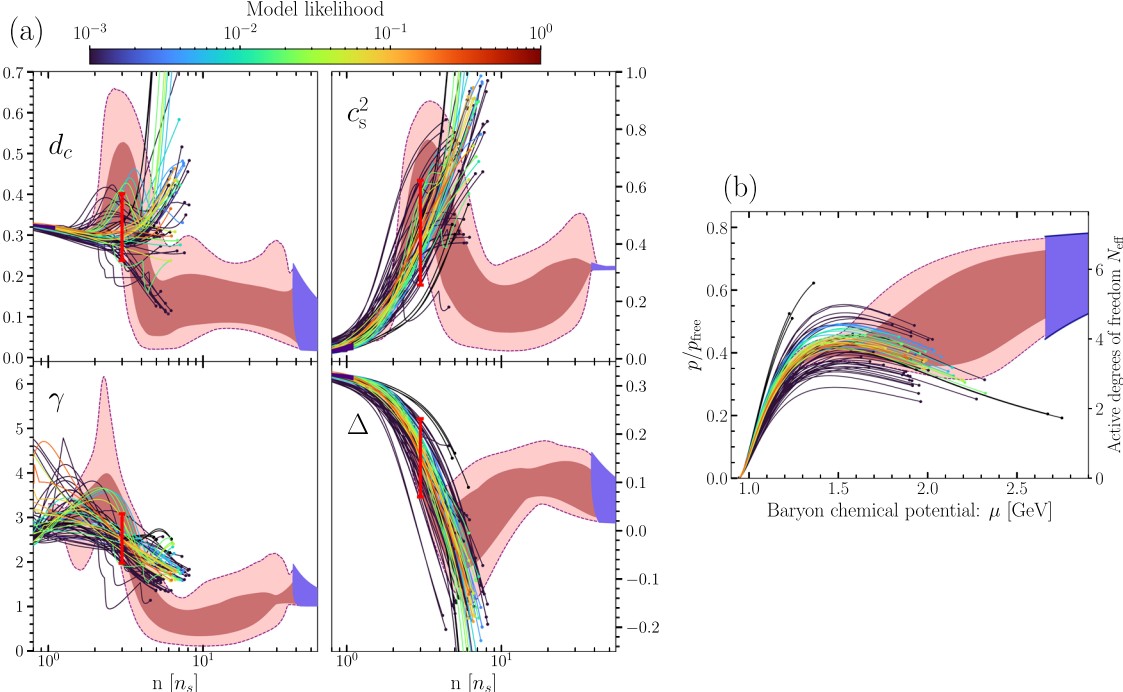

**Fig. 8 | Comparison to nuclear-matter models. a** The physical quantities displayed in Figs. 1, 2, but this time including as the thin lines all purely hadronic ($T = 0$, $\beta$-equilibrium) models available on the CompOSE database[90]. The coloring of the individual EoSs corresponds to the relative likelihoods of the models in comparison with the most likely EoS from the GP ensemble. This likelihood depends only on the astrophysical and high-density pQCD inputs. The pQCD constraint is imposed at the TOV points where these curves end, and is implemented by testing whether these points can be connected to the pQCD limit at $40n_s$ in a causal and thermodynamically consistent fashion. The solid red vertical lines here represent the ranges displayed in Table 1, and the dark and light red regions in the background correspond to our $c_{s,4}^2$ interpolation results. **b** The normalized pressure of Fig. 4, but including now also the purely hadronic ($T = 0$, $\beta$-equilibrium) model EoSs displayed in panel a. The curves are again set to end at the respective TOV points.

figure we display the behavior of the normalized pressure in the same NM models, noting that in most cases the values of the quantity are undergoing a slow decline at the TOV point. This appears to be at odds with the general trend of the interpolated EoS band.

## Data availability

Thinned versions of the EoS ensembles generated for this study (interpolations and GP) have been deposited to the Zenodo data repository and are available separately for the interpolations[92] and the GP method[93].

## Code availability

All codes written for the generation and analysis of the EoS ensembles are available from the authors upon request.

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

## Acknowledgements

We would like to thank Kenji Fukushima, Sophia Han, Carlos Hoyos, Niko Jokela, and Anna Watts for useful discussions. The work has been supported by the Finnish Cultural Foundation (EA), the Academy of Finland grants no. 1303622 and 354533 (EA, JH, and AV), the European Research Council grant no. 725369 (EA, JH, and AV), the Deutsche Forschungsgemeinschaft (DFG, German Research Foundation) – project-ID 279384907 – SFB 1245 (TG), the State of Hesse within the Research Cluster ELEMENTS, project-ID 500/10.006 (TG), and a joint Columbia University/Flatiron Institute Research Fellowship (JN). The authors also acknowledge CSC - IT Center for Science, Finland, for computational resources (project 2003485).

## Author contributions

We are listed in alphabetical order. All authors (E.A., T.G., J.H., O.K., A.K., J.N., A.V.) conducted the research together and participated in the preparation and revision of the manuscript. The GP framework was originally developed by O.K. and A.K. and implemented here by O.K. and T.G.. The MCMC interpolation framework was originally developed by J.N. and implemented here by J.H., E.A., and J.N.

## Competing interests

The authors declare no competing interests.
