## [Peer Review File · Nature Communications]

Strongly interacting matter exhibits deconfined behavior in massive neutron starsREVIEWER COMMENTS

Reviewer #1 (Remarks to the Author):

In the present study the authors propose indicators that quantify the possible existence of deconfined quark matter inside a neutron star. This is achieved starting from the low density and high density equation of state of strongly interacting matter obtained from ab-initio calculations. The intermediate densities are spanned by three different agnostic approaches (polytropic, speed of sound and gaussian processes) using Bayesian inference. Several observational data, and in the case of GP also the pQCD constraints, are used to calculate the final probability distributions. The authors propose a new quantity, dc , related with the trace anomaly to define the probability of restoration of the conformal symmetry. It is proposed that a value below 0.2 indicates conformal symmetry. Taking this quantity as indicative, together with the value of p/p_{free} , they find that in the core of the most massive neutron stars quarks are most likely deconfined. The confirmation of the existence of deconfined quark matter inside massive stars, and identification of signatures for its occurrence is one of the main quests that make neutron stars as possible important nuclear and particle physics laboratories, and, therefore, the main objective of the study is of importance for the field. However, the following points need clarification before a final recommendation can be drawn:

- in Table 1, Dense NM ("referring to nuclear-matter model predictions at densities corresponding to cores of massive NS close M_{TOV} ") is characterized by properties that should be explained. The authors indicate reference 3 that does not study explicitly this kind of matter. A recent publication of EoS that have been made public in CompOSE [Malik & Pais (Eur. Phys. J. A (2022) 58:154)] gives the EOS and the properties of a set of eight quite different EOS based on a relativistic mean-field description and proposed by different research groups. In Table 2 of this publication, four of these models have cs^2 below 0.5 and only one reaches 0.8 at the centre of the star with a mass equal to M_{TOV} . In this paper quantities such as γ and dc are not presented but they can be easily calculated from the CompOSE dataset and it is easy to see that, for models such as TM1e and FSU2R with a speed of sound squared the order of 0.4, dc goes below 0.2 at densities above four times saturation density, and γ goes below 1.7 below three times saturation density. The

high density behaviour of the speed of sound in RMF and the discussion under which conditions these models predict a speed of sound square $1/3$ in the high density limit was presented in Mueller nucl-th/9603037, and recently these properties were explored within a Bayesian inference description in Malik arxiv:2301.08169. These authors have shown that nuclear models may predict sound speeds in the center of massive stars below $0.5c^2$. Taking values in the range $0.8-1c^2$ as the speed of sound at the center of the M_{TOV} star seems to indicate the authors have considered non-relativistic models.

- Another argument of the authors in favour of the identification of deconfined matter is related to the behaviour of the ratio p/p_{free} . The authors obtain a flattening of the normalized pressure with a value of 0.4 ± 0.03 at 68% CI corresponding, at the same level of confidence to $N_{\text{eff}} = 3.6 \pm 0.3$. However, RMF models present the same flattening and for value of the ratio $p/p_{\text{free}} \sim 0.35$ and $N_{\text{eff}} > 3$ [results obtained using models of Malik & Pais (Eur. Phys. J. A (2022) 58:154)]. Could you comment?

- The authors refer that they want to "distinguish certain local behaviours of the EoS, such as the onset of hyperonic degrees of freedom". Could the authors clarify how is this possible in their approach since the hyperon onset (or other non-nucleonic degrees of freedom such as the Delta-resonance or kaon condensation) comes generally as a smooth crossover, not different from the expected behaviour of the EoS constructed from several segments. An hyperon onset brings γ below 1.7.

- in ref. 27 considering a SMNS scenario, the authors have predicted a mass $M_{\text{TOV}} < 2.53 M_{\text{Sun}}$. Is it clear why in the present approach the same ansatz gives $M_{\text{TOV}} = 2.27 \pm 0.11 M_{\text{Sun}}$?

- the authors have used four segments in the speed of sound description arguing that this is enough. In ref 29, the authors show that five segments should be the minimum that covers all the allowed phase space in the pressure-energy density plane, although no Bayesian inference was applied. The effect of taking 4 instead of 5 segments shown in fig. 5 is equivalent to the one discussed in [29]: the spread of the data is larger if calculated with 5 segments. In particular, while with 4 segments dc is well below 0.2 above $n > \sim 6n_{\text{sat}}$, with 5

segments the probability of getting values of dc above 0.2 is larger. This should be commented.

- the authors have checked their results with three different approaches to the agnostic description of the EoS. At high densities the GP approach seems to spread over a much wider range of values than cs^4 , even after pQCD constraints are imposed, for instance, cs^2 attains 1, and dc 68% CI goes up to 0.3 above the M_{TOV} central density. Besides it is indicated the $\gamma < 1.75$ gives a 99.8% probability of conformalization taking the speed of sound approach. How much would this probability decrease using the GP description? Could you comment? The left and right middle panels of figure 2 do not seem to be consistent: close to M_{TOV} GP follows the 95% CI of cs^4 in the left panel but coincides with the 68% CI in the right panel.

- In fig 1, why is the $2M_{sun}$ band not extending above 0.2?

- in eq (A1) and following discussion, you introduce a parameter γ different from the one of the main paper. This should be changed. This parameter is introduced to allow the incompressibility to vary. Which values of K characterize the generated EoS? Are the GP EoS generated imposing the same low density constraints?

In summary, I agree that quantities as the trace anomaly, dc indicate conformalization of matter but not necessarily deconfinement, as the authors also state. To overcome this problem the authors suggest that the value of p/p_{free} which is proportional to the number of degrees of freedom could bring the lacking information and consider that the value of $N_{eff} = 3.6 \pm 0.3$ is indicative of a growing number of degrees of freedom. However, the results obtained within a hadronic description do not seem to be very different.

Reviewer #2 (Remarks to the Author):

"Strongly interacting matter exhibits deconfined behavior in massive neutron stars" by Annala et al.

Summary: The authors of this paper investigate the likelihood of quark deconfinement in the cores of neutron stars by combining information from astrophysical observations (neutron star masses, radii, tidal deformabilities) and theoretical calculations. Using Bayesian inference, they find that in the cores of the most massive neutron stars, the equation of state (EoS) is consistent with quark matter. This finding rests on two key aspects: (i) the establishment of approximate conformal symmetry restoration (88% level of confidence at the highest densities observed in neutron stars), and (ii) the demonstration that the count of active degrees of freedom aligns favorably with an interpretation of this observation as deconfined matter. The latter point holds significant importance as the establishment of conformalization of matter does not inherently indicate quark deconfinement.

Main results: Having presented evidence through the analysis of various properties, including the speed of sound, the polytropic index, the normalized trace anomaly, and its logarithmic derivative, the authors proceed to investigate the active degrees of freedom within such matter. Based on their findings (e.g., the ratio p/p_{free} is approximately two-thirds of its value in weakly coupled quark matter), they conclude that the existence of quark matter in the cores of the most massive neutron stars is highly probable. This finding is derived from meticulous statistical investigations. The results are thoroughly discussed and analyzed in the Letter.

Comments, Criticisms, and Suggestions:

- It is of utmost importance to stress that the conclusion regarding the existence of quark matter in the cores of the most massive neutron stars is based solely on equations of state that do not consider the

potential occurrence of a first-order phase transition within the neutron star cores. While this aspect is briefly mentioned in the Letter, it is essential to clearly emphasize it in the abstract and (possibly rephrase the) title of the paper. By doing so, readers who are less familiar with this specific research topic can avoid drawing inaccurate conclusions, especially in relation to the findings presented in Reference [70].

- The statement "At high densities and low temperatures, the situation is in many ways similar to the above" (page 5) is just something that is said in passing. A brief discussion should be added since heavy-ion matter differs significantly from dense neutron star matter.

- In the Abstract, the authors write "This highly compressed matter may undergo a phase transition where neutron-rich nuclear matter melts into deconfined quark matter, liberating its constituent quarks and gluons [1]", where [1] is Shuryak's 1980 Phys. Rept. paper.

The authors are likely aware that the notion of the possible existence of quark matter in the cores of neutron stars was not initially proposed by Shuryak. Prior to his work, this idea had been put forth by:

Ivanenko, Dmitri D., and Kurdgelaidze, D. F. (1965) "Hypothesis concerning quark stars", *Astrophysics* 1 (4), 251–252

Ivanenko, Dmitri D.; Kurdgelaidze, D. F. (1969). "Remarks on quark stars", *Lettere al Nuovo Cimento* 2, 13–16

H. Fritzsch, M. Gell–Mann, and H. Leutwyler, *Phys. Lett.* 47B (1973) 365.

- Is there a specific motive behind the authors' choice to display the pressure as a function of the baryon chemical potential in Figure 4, instead of baryon number or energy density? By presenting pressure as a function of either baryon number or energy density, it would establish a more direct connection for many readers to the densities observed in the cores of neutron stars. This choice would likely be more illustrative and enhance the readers' comprehension.

- In Figures 5-7, it would be beneficial to indicate the central densities achieved in at least some of the M-R models in the corresponding pressure versus energy density plots.

Recommendation: I believe this paper makes a significant contribution to the current literature on the potential presence of quark matter in neutron stars. However, in order for me to confidently recommend their work for publication, it is necessary for the authors to address the concerns and clarify the points I have raised above.

Reviewer #3 (Remarks to the Author):

The authors employ the assumption that at extremely high densities, more than 7 times the central density of any neutron star, the equation of state (EOS) can be predicted from perturbative-QCD (pQCD) calculations. Furthermore, the authors claim that this constraint on the equation of state constrains the equation of state within neutron stars. They have made this claim in the literature previously, for example Annala et al., Phys. Rev. Lett. 120, 172703 (2018); arXiv.1904.01354 (2019); Nat. Phys. 16, 907 (2020); Phys. Rev. X 12, 011058 (2022). If true, this represents an important consideration for modeling neutron stars. This paper is an extension of previous work by the group, in which the properties of pQCD matter is evaluated and methods of interpolating through the large density range from pQCD validity to neutron star densities are considered.

The authors claim their analysis is able to show that matter at the highest densities in

neutron stars is consistent with that of quark matter. They further state their analysis demonstrates that quark matter most likely exists at the highest densities found in the most massive neutron stars. The latter is based on a statistical sampling of model equations of state (both parameterized and so-called non-parametric equations of state) constrained at low densities by chiral effective field theory calculations of neutron matter and at higher densities by observed properties of neutron stars.

As far as I can determine, the authors' results follow from their various assumptions, but I think that more consideration of the effects of these assumptions is warranted.

The authors state that with few exceptions, model-agnostic studies of the neutron star matter EOS fail to take into account information from pQCD calculations. I don't think this omission is as serious as the authors suggest. All of these studies employ the constraint that matter remain causal at all densities. The squared speed of sound c_s^2 can be shown to approximate or exceed $1/3$ at densities around $3n_s$ (nuclear saturation density), which is about half the central density of the most massive stars, otherwise the EOSs won't be able to support 2 solar masses or more. At this approximate central density, stars typically have masses slightly larger than 1.4 solar masses (see Figure 1, for example). The properties of neutron stars with larger central densities and masses are relatively insensitive to the EOS at higher densities, as long as it remains causal and as long as 2 solar mass stars are able to be supported. Certainly, the properties of 1.4 solar mass neutron stars, the most likely value for X-ray sources and merger components (which provide most of the available astrophysical information concerning radii), are completely determined by the EOS below this density. But studies have shown that even the properties of 2 solar mass stars are little influenced by the EOS above this density (unless a strong first-order phase transition appears just above this density). Therefore, whether or not the squared sound-speed limit above $40n_s$ is taken to be 1 or $1/3$ is not likely to be important. And at high densities, if c_s^2 is approximately unity, by implication γ becomes approximately 2, only slightly above the pQCD value.

The authors need to make the case that imposition of the pQCD

constraint actually does impact things like the radii of normal (1.3-1.6 solar mass) neutron stars as well as the maximum neutron star mass before claiming that "most likely" quark matter exists in the most massive neutron stars. One way to show this would be to make a test calculation in which pQCD was replaced by a sound speed of unity and, by implication, $\gamma=2$, (at the same high densities equal to and exceeding $40n_s$), and employing the same astrophysical constraints. If the predictions of radii (or, equivalently, tidal deformabilities or moments of inertia) are thereby significantly affected, the authors' case will have been made.

Actually, a telling calculation would be to assume somewhat the opposite: remove all astrophysical constraints with the exception of a 2 solar mass lower limit to the maximum mass, but keep the low-density chiral EFT, causality, and the high-density pQCD constraints. What would the new preferred EOS and M-R bands compare to the ones in which other astrophysical constraints were kept? If the bands are very similar, this would seriously weaken the authors' conclusions.

Another problem with the manuscript is that the authors do not show a mass-radius diagram, which is a standard representation for neutron stars. How does their preferred most-likely band compare to what is obtained by assuming the ultimate sound speed is unity instead of $1/3$?

The role of astrophysical constraints is a little ambiguous. X-ray observations, in particular, are subject to large systematic uncertainties, especially those from quiescent and bursting sources where the distance uncertainties are important. If each observational constraint (with uncertainties) were shown in an M-R diagram, the reader could assess the significance of each constraint. One would have to include distance uncertainties in both quiescent and bursting sources, unlike what is often shown in the cited papers, which frequently assume a distance. Ideally, the significance of each type of observational constraint (pulsar mass measurements, tidal deformabilities, maximum mass upper limit from GW170817, NICER, bursting sources, and quiescent sources) would be delineated.

How important is each observational constraint to localizing the most-likely band in either the M-R diagram or the pressure-energy density diagram? And how does each observational constraint affect the likelihood of quark matter in the densest stars?

In conclusion, the authors need to back up their claim that including the pQCD constraint is essential to assessing the probability of a quark-hadron transition in the densest neutron stars. Replacing the pQCD constraint with another arbitrary constraint (such as a high-density sound speed limit of 1 or $2/3$) should suffice. The role of astrophysical constraints (apart from that of imposing a minimum maximum mass) should also be clarified; their importance might be assessed from a calculation in which they are removed. Once these tests have been completed, this paper could be a valuable contribution to the study of quark matter in neutron stars. None of their previous papers have attempted such comparisons.

Reply to referee 1:

We sincerely thank Referee 1 for their overall positive evaluation of our work and in particular for their extremely useful comments concerning nuclear matter (NM) model EoSs. These comments prompted us to perform several new analyses, which we believe led to a significant sharpening of our argument.

Referee: In the present study the authors propose indicators that quantify the possible existence of deconfined quark matter inside a neutron star. This is achieved starting from the low density and high density equation of state of strongly interacting matter obtained from ab-initio calculations. The intermediate densities are spanned by three different agnostic approaches (polytropic, speed of sound and gaussian processes) using Bayesian inference. Several observational data, and in the case of GP also the pQCD constraints, are used to calculate the final probability distributions. The authors propose a new quantity, d_c , related with the trace anomaly to define the probability of restoration of the conformal symmetry. It is proposed that a value below 0.2 indicates conformal symmetry. Taking this quantity as indicative, together with the value of p/p_{free} , they find that in the core of the most massive neutron stars quarks are most likely deconfined. The confirmation of the existence of deconfined quark matter inside massive stars, and identification of signatures for its occurrence is one of the main quests that make neutron stars as possible important nuclear and particle physics laboratories, and, therefore, the main objective of the study is of importance for the field. However, the following points need clarification before a final recommendation can be drawn:

Our reply: We thank the referee for their encouraging comments and naturally agree on the importance of the problem addressed in our work. We emphasize that the pQCD constraint is part of our setup in all three independent implementations, with the main difference between the GP and interpolation methods being that in the GP calculation it can be turned on and off at will whereas in the interpolation method, it is always present by construction.

Referee: In Table 1, Dense NM (“referring to nuclear-matter model predictions at densities corresponding to cores of massive NS close M_{TOV} ”) is characterized by properties that should be explained. The authors indicate reference 3 that does not study explicitly this kind of matter. A recent publication of EoS that have been made public in CompOSE [Malik & Pais (Eur. Phys. J. A (2022) 58:154)] gives the EOS and the properties of a set of eight quite different EOS based on a relativistic mean-field description and proposed by different research groups. In Table 2 of this publication, four of these models have c_s^2 below 0.5 and only one reaches 0.8 at the centre of the star with a mass equal to M_{TOV} . In this paper quantities such as γ and d_c are not presented but they can be easily calculated from the CompOSE dataset and it is easy to see that, for models such as TM1e and FSU2R with a speed of sound squared the order of 0.4, d_c goes below 0.2 at densities above four times saturation density, and γ goes below 1.7 below three times saturation density. The high density behaviour of the speed of sound in RMF and the discussion under which conditions these models predict a speed of sound square 1/3 in the high density limit was presented in Mueller nucl-th/9603037, and recently these properties were explored within a Bayesian inference description in Malik arxiv:2301.08169. These authors have shown that nuclear models may predict sound speeds in the center of massive stars below 0.5. Taking values in the range 0.8-1 as the speed of sound at the center of the M_{TOV} star seems to indicate the

Figure 1: The physical quantities plotted in figs. 1-2 of the manuscript, but now displaying as the thin lines the ($T = 0$, β -equilibrium) EoSs of all nuclear matter models available on the CompOSE database. The models that include a transition to quark matter are not shown here, as the purpose of the figure is to make a comparison to fully hadronic stars. The coloring of the individual EoSs corresponds to the posterior likelihoods in our analysis, including all astrophysical observations discussed in the paper. The pQCD constraint is imposed at the TOV point where these curves end, and is implemented by testing whether these points can be causally and consistently connected to the pQCD limit. The posterior likelihoods are normalized by the highest likelihood obtained in our GP ensemble, while the solid red vertical lines represent the ranges of these quantities included in the table I of the manuscript.

authors have considered non-relativistic models.”

Another argument of the authors in favour of the identification of deconfined matter is related to the behaviour of the ratio p/p_{free} . The authors obtain a flattening of the normalized pressure with a value of 0.4 ± 0.03 at 68% CI corresponding, at the same level of confidence to $N_{eff} = 3.6 \pm 0.3$. However, RMF models present the same flattening and for value of the ratio $p/p_{free} \sim 0.35$ and $N_{eff} > 3$ [results obtained using models of Malik & Pais (Eur. Phys. J. A (2022) 58:154)]. Could you comment?

Our reply: We agree with the referee that our handling of the nuclear matter EoSs was not fully up to date in the first version of our draft, which relied on an earlier analysis performed for our 2020 Nature Physics article. As the referee points out, several important EoSs were missing from this set, some (but not all) of which have been derived since 2020. To this end, we have now significantly extended our analysis by downloading all available hadronic $T = 0$ EoSs from the CompOSE database and examining their properties, including the conformal distance d_c and pressure p . Importantly, the

set of EoSs we have now analyzed includes all the individual EoSs that Referee 1 mentions in their report. We sincerely thank the referee for bringing these important EoSs to our attention.

The main findings of our new analysis are displayed in the above fig. 1, the left panel of which is also included in the new version of our manuscript. The corresponding results can be summarized in the following set of observations:

1. For most quantities studied, there is overall agreement between the different NM EoSs only up to densities of order $3n_n$, roughly corresponding to the central densities of typical pulsars, but significant deviations begin soon thereafter. As long as there is agreement, the EoSs exhibit highly nonconformal properties.
2. Proceeding to higher densities, where the validity of the models becomes questionable due to the emergence of new resonances as well as unknown interactions between them, a small fraction of the models exhibits near-conformal properties. Indeed, there are even a handful of individual EoSs for which $d_c < 0.2$ at TOV densities, which seemingly provides a counterexample to our main claim given that these are by construction hadronic models.
3. Upon closer inspection, *all models consistent with near-conformal matter at TOV densities are, however, seen to have negligibly small likelihoods in our analysis*, which can be traced back to a tension with the X-ray measurements included in our analysis.
4. The referee was absolutely right in saying that there are several nuclear matter model EoSs, for which the normalized pressure is comparable to our results. Moreover, there are even multiple models that carry meaningful likelihoods in our analysis while doing so. The normalized pressure does not stabilize in these models, though, but for most EoSs is already on its way down at the TOV density, which is in stark contrast to the behavior we witness in our model-agnostic calculation.

To make the discussion in the new draft consistent with the above findings, we have now significantly updated our discussion concerning the nuclear matter model EoSs:

- We have redefined the "Dense NM" column of Table I to not correspond to the central densities of TOV stars (where several entries would have enormous ranges), but rather to the highest density ($\approx 3n_s$) where there is at least qualitative agreement between different NM models. We have used the models plotted in the above fig. 1 to determine the ranges that are now indicated in Table I.
- We have now significantly extended our discussion of the NM model EoSs in the main text (see p. 2 and 4 and the new footnote 2) and in addition written an entirely new subsection F to the Methods section where we discuss the above fig. 1 (left), now included as the new fig. 5 of the modified draft.

Referee: The authors refer that they want to "distinguish certain local behaviours of the EoS, such as the onset of hyperonic degrees of freedom". Could the authors clarify how is this possible in

their approach since the hyperon onset (or other non-nucleonic degrees of freedom such as the Delta-resonance or kaon condensation) comes generally as a smooth crossover, not different from the expected behaviour of the EoS constructed from several segments. An hyperon onset brings gamma below 1.7.

Our reply: It is of course true that an approach such as ours, which studies the EoS alone, cannot fully distinguish between different microscopic sources of the observed EoS behaviors. However, as the analysis presented in fig. 1 above reveals, no hadronic EoS (including all the hyperonic ones) is able to reproduce the observed near-conformal behavior at TOV densities while respecting all observational constraints, which gives us confidence in our interpretation of the observed conformalization.

What we referred to as "local behaviors" in our initial manuscript was features where an EoS briefly becomes near-conformal but then returns to non-conformality; some examples of such behavior are visible in the model predictions for d_c and γ in the above fig. 1 (left) both slightly below and above the vertical line at $3n_s$. In contrast, the kind of conformalization we associate with the deconfinement transition is one where d_c and other indicators of conformality remain safely conformal to higher densities.

In order to clarify our statement and avoid suggesting that we can confidently rule out certain microscopic mechanisms behind the observed EoS behavior, we have now removed any reference to "local behaviors" on p. 2 and instead merely state that to ensure the approximate conformalization of the system at a given density, one must track several quantities and ensure that the system does not return to non-conformal behavior at any higher density.

Referee: In ref. 27 considering a SMNS scenario, the authors have predicted a mass $M_{\text{TOV}} < 2.53M_\odot$. Is it clear why in the present approach the same ansatz gives $M_{\text{TOV}} = 2.27 \pm 0.11M_\odot$?

Our reply: The difference between the quoted M_{TOV} values mostly arises from the fact that the former (larger) value originated from a "hard-cut" analysis while the present calculation is performed in a Bayesian framework. In particular, the value $M_{\text{TOV}} < 2.53M_\odot$ from our earlier PRX article corresponds to the maximal value M_{TOV} can reach assuming that the corresponding EoS satisfies the 90% confidence limits of a handful of neutron-star measurements. Here we, on the other hand, properly take into account the observational uncertainties of a (larger) set of measurements as well as the marginalization over prior distributions. This leads us to the quoted 68% credible region $M_{\text{TOV}} = 2.27 \pm 0.11M_\odot$ as well as somewhat wider 90% and 95% credible regions.

Referee: The authors have used four segments in the speed of sound description arguing that this is enough. In ref 29, the authors show that five segments should be the minimum that covers all the allowed phase space in the pressure-energy density plane, although no Bayesian inference was applied. The effect of taking 4 instead of 5 segments shown in fig. 5 is equivalent to the one discussed in [29]: the spread of the data is larger if calculated with 5 segments. In particular, while with 4 segments d_c is well below 0.2 above $n \gtrsim 6n_s$, with 5 segments the probability of getting values of d_c above 0.2 is larger. This should be commented.

Our reply: This is admittedly a nontrivial point and one that we had been discussing at length already during the preparation of the initial manuscript. Below, we first explain the motivation behind our initial choice and then discuss possible alternative strategies.

First of all, it is important to point out that the use of interpolation ansätze with a very high number of segments leads to a rapid increase in the computational cost of the runs in this type of a Bayesian calculation. For instance, a full analysis with the five-segment cs5 parametrization would take tens of millions of CPU hours to successfully complete, while with the cs4 method a few million CPU hours suffices. Given the cost and in particular duration of the computer runs, this is an important factor to consider and the cs5 method should only be used if deemed necessary.

Second, our past experience with different interpolators in hard-cut (i.e. non-Bayesian) studies such as 2105.05132 indicates that the difference between four- and five-segment speed-of-sound ansätze is typically very small. This difference should be even more suppressed in a Bayesian calculation, where the most extreme EoSs (that require a high number of parameters) typically carry very small probabilities. While this may indeed be somewhat different for polytropes (as the referee remarks), it is good to recall that we use the four-segment polytrope setup only as a qualitative "sanity check" and not as one of our main results.

Having said the above, it is of course true — as the referee points out — that there are noticeable (albeit minor) differences between the cs4 and cs5 results in our fig. 5. While we suspect them to be largely due to the incompleteness of the cs5 run and consider the evolution of the QM likelihoods with increasing number of segments ($95 \pm 2\% \rightarrow 88 \pm 2\% \rightarrow 84\% \pm 5\%$ for cs3 \rightarrow cs4 \rightarrow cs5) a sign of good convergence, we perfectly understand the referee's wish for further confirmation. To address these concerns, we have now performed the following re-analyses and changes in the manuscript:

- We successfully applied for more computing resources in June and immediately started a new cs5 run, the results of which have, however, not appreciably changed from the ones reported in our first version. In particular, the likelihood of QM cores from the cs5 run is still the same as before, only the uncertainty has slightly reduced; this is in line with the fact that a reduction of the uncertainty by a factor of 2 typically requires an order of magnitude more samples. A jack-knife analysis indicates that the posterior evidence values still have large fluctuations of the order of $\pm 5\%$, so we can merely conclude that the cs5 calculations offer qualitative support for our cs4 results but do not warrant a full quantitative assessment on the same level of confidence as the latter model (for cs4 same analysis yields $\pm 2\%$). For clarity, we have now changed the cs5 panel in fig. 5 to read "cs5 (preliminary)" to highlight the incompleteness of this run.
- We have now lifted the GP method likelihood for QM cores (75%) on par with the cs4 likelihood, removing the 88% figure from our abstract and reporting both in a similar fashion on p. 4 (see also our reply to the following comment). As we note in the manuscript, it is good to recall that the GP implementation uses the pQCD limit in a considerably less constraining fashion than the interpolation method and allows the most general EoS behavior possible, so that the corresponding QM likelihood represents a lower limit for this quantity.

We hope that these changes suffice to alleviate the referee's justified concerns in this matter.

Referee: The authors have checked their results with three different approaches to the agnostic description of the EoS. At high densities the GP approach seems to spread over a much wider range of values than cs4, even after pQCD constraints are imposed, for instance, c_s^2 attains 1, and d_c 68% CI

goes up to 0.3 above the M_{TOV} central density. Besides it is indicated the $\gamma < 1.75$ gives a 99.8% probability of conformalization taking the speed of sound approach. How much would this probability decrease using the GP description? Could you comment?

Our reply: The behavior described by the referee arises from the extremely conservative way the pQCD input is taken into account in the GP setup (see also our answer to the previous point). In this framework, we simply test, whether the last point of the GP EoS at $10n_s$ can be connected to the pQCD EoS at high density ($\mu_B = 2.6\text{GeV}$ or $n \sim 40n_s$) with a stable and causal function. Unlike in the case of the c_s^2 interpolation, this makes no use of additional information on the EoS at these intermediate densities and in addition allows for arbitrarily complicated EoS behavior. The EoS is also not marginalized over the density range $[10n_s, 40n_s]$, and as a result the uncertainties of local quantities such as c_s^2 grow rapidly around the density where the pQCD constraint is applied. For these reasons, we presented the cs4 interpolation as our main result in the first version of the manuscript and used the GP results merely as a (very conservative) comparison point, providing a consistency check¹ for our results and a lower limit for the QM likelihood.

As the referee points out, the GP results indeed significantly broaden near the matching point of $10n_s$, which is particularly clear in the speed of sound. This is because the pQCD input does not directly constrain c_s^2 but rather its (double) integral moments, i.e. the triplet ϵ , p , and n . As expected, the broadening is significantly less visible in $p(\epsilon)$ (see the bottom right column of Fig. 7).

Finally, we note that we have now computed the QM probability using the $\gamma < 1.75$ criterion also with GP method, obtaining the result 97.8% which we also quote in the manuscript. Note that while this figure is very high as well, the remaining likelihood for purely hadronic TOV stars (2.2%) is 11 times larger than for the cs4 interpolation, reflecting the above-discussed differences between the two frameworks.

Referee: The left and right middle panels of figure 2 do not seem to be consistent: close to M_{TOV} GP follows the 95% CI of cs4 in the left panel but coincides with the 68% CI in the right panel.

Our reply: We have checked this issue, and both the left and right middle row plots are consistent and correct. The apparent inconsistency appears to arise from the EoS-dependent (and very non-trivial) relation between the central density and NS mass.

Referee: In fig 1, why is the $2M_{\odot}$ band not extending above 0.2?

Our reply: This is purely a graphical design choice made to reduce clutter in the figure. Note that the heights of these vertical bars do not carry any physical information, but the sole purpose of the bars is to indicate the (horizontal) locations of the central densities.

Referee: In eq (A1) and following discussion, you introduce a parameter γ different from the one of the main paper. This should be changed. This parameter is introduced to allow the incompressibility to vary. Which values of K characterize the generated EoS? Are the GP EoS generated imposing the same low density constraints?

¹Had the GP method led to a qualitatively differing result for some physical quantity, this would have called for additional scrutiny.

Our reply: We thank the referee for pointing out this unfortunate notational choice of ours. Indeed, in the appendix we followed the notation of 1303.4662 which conflicts with our own definition of γ . We have now changed to using capital Γ for the low-energy parameter in question, and in addition note that the EoSs we use correspond to values of the K parameter between 100 and 250 MeV. While the lower limit may appear rather small, we note that the resulting EoS appears to be highly insensitive to the values that K takes.

Finally, the GP construction follows the procedure described in 2204.11877, where the handling of the low-density CEFT EoS differs slightly from that used for our interpolated EoSs. The numerical differences are, however, very small, and the fact that our final results agree to such a good extent gives us confidence in our conclusions.

Referee: In summary, I agree that quantities as the trace anomaly, d_c indicate conformalization of matter but not necessarily deconfinement, as the authors also state. To overcome this problem the authors suggest that the value of p/p_{free} which is proportional to the number of degrees of freedom could bring the lacking information and consider that the value of $N_{eff} = 3.6 \pm 0.3$ is indicative of a growing number of degrees of freedom. However, the results obtained within a hadronic description do not seem to be very different.

Our reply: We agree with the referee that the difference between conformalization and deconfinement is an important issue. As commented above, our analysis concerning the number of effective degrees of freedom does not alone represent conclusive evidence for the latter but is merely one quantity pointing in this direction. It is also worth pointing out that while several nuclear-matter models indeed predict comparable values for p/p_{free} at the centers of maximal-mass NSs, for these EoSs this quantity is already decreasing, having reached a maximum at somewhat smaller densities. This is in stark contrast to our model-agnostic EoSs, for which the number of effective degrees of freedom keeps slowly increasing towards the perturbative region. The latter behavior is precisely what is observed in high- T quark-gluon plasma and what is expected for high-density quark matter.

Reply to referee 2:

We thank referee 2 for their valuable and generally positive comments and suggestions. We agree on most of the points raised, and are particularly grateful for the suggested references concerning the appearance of deconfined matter inside compact stars. Below, we comment on the various points raised, paying particular attention to the one concerning first-order phase transitions.

Referee: Summary: The authors of this paper investigate the likelihood of quark deconfinement in the cores of neutron stars by combining information from astrophysical observations (neutron star masses, radii, tidal deformabilities) and theoretical calculations. Using Bayesian inference, they find that in the cores of the most massive neutron stars, the equation of state (EoS) is consistent with quark matter. This finding rests on two key aspects: (i) the establishment of approximate conformal symmetry restoration (88% level of confidence at the highest densities observed in neutron stars), and (ii) the demonstration that the count of active degrees of freedom aligns favorably with an interpretation of this observation as deconfined matter. The latter point holds significant importance as the establishment of conformalization of matter does not inherently indicate quark deconfinement.

Main results: Having presented evidence through the analysis of various properties, including the speed of sound, the polytropic index, the normalized trace anomaly, and its logarithmic derivative, the authors proceed to investigate the active degrees of freedom within such matter. Based on their findings (e.g., the ratio p/p_{free} is approximately two-thirds of its value in weakly coupled quark matter), they conclude that the existence of quark matter in the cores of the most massive neutron stars is highly probable. This finding is derived from meticulous statistical investigations. The results are thoroughly discussed and analyzed in the Letter.

Our reply: We agree on the referee's summary of our paper and its main results and appreciate their kind evaluation of the quality of our work.

Referee: It is of utmost importance to stress that the conclusion regarding the existence of quark matter in the cores of the most massive neutron stars is based solely on equations of state that do not consider the potential occurrence of a first-order phase transition within the neutron star cores. While this aspect is briefly mentioned in the Letter, it is essential to clearly emphasize it in the abstract and (possibly rephrase the) title of the paper. By doing so, readers who are less familiar with this specific research topic can avoid drawing inaccurate conclusions, especially in relation to the findings presented in Reference [70].

Our reply: The possibility of a discontinuous first-order phase transition is indeed an important issue, and we agree that our discussion on this topic was too brief in the first version of our manuscript. We have now extended it both in the abstract and main text (see below for details), but would in addition like to note that there are strong reasons to expect that our main findings will remain unchanged upon the addition of explicitly discontinuous transitions to our framework (ongoing work by Oleg Komoltsev). The reasons for this are twofold:

1. Both our c_s^2 interpolation and GP approaches do a good job in mimicking mild and moderate first-order transitions. For the GP approach, this question was recently addressed by Essick et al. in 2305.07411 with a positive conclusion, while for the cs4 interpolation, discussed in our

earlier Nature Physics and PRX articles 1903.09121 and 2105.05132, the same effect is achieved by allowing the speed of sound to remain arbitrarily close to zero for extended density intervals. As mentioned in our manuscript, for the cs4 interpolation we were in fact able to identify that the 12% probability of purely hadronic TOV stars is associated to first-order-PT-like behavior, with the $M-R$ curve ending when $c_s^2 \approx 0$. The implementation of even a small artificial lower limit for c_s^2 or an upper limit for the length of the intervals where the quantity remains small would have the effect of elevating the likelihood of QM cores in TOV stars to nearly 100%.

2. As noted by the referee, in ref. [70] a subset of us indeed found that by allowing sizable discontinuous phase transitions beginning at a very low density, it is possible to construct EoSs that extend beyond hard-cut results derived with approximate PTs. Here, it is however worth noting, that (i) very few such EoSs were found in the posterior of that work (ii) all such EoSs were found to lie right at the boundary of the allowed parameter space. Hence, they carried small posterior weights in the approach taken in [70], and we strongly suspect that further suppression will take place once the full measurement uncertainties are included. We thus conclude that the likelihood estimates for QM cores we have currently obtained will likely experience only very minor shifts with the addition of explicitly discontinuous transitions.

Having said the above, we agree with the referee that the limitations of our current approach should be further highlighted. To this end, we have extended our discussion in two places: in the abstract, we now mention the approximate description of phase transitions in the last sentence, and on p. 5 of the main text, we now argue in more detail (along the above lines) why we do not believe that our results will appreciably change upon the eventual addition of explicit PTs.

Referee: The statement "At high densities and low temperatures, the situation is in many ways similar to the above" (page 5) is just something that is said in passing. A brief discussion should be added since heavy-ion matter differs significantly from dense neutron star matter.

Our reply: We thank the referee for pointing out this somewhat loose language used in the first version of the manuscript. We have now reformulated this sentence to note that the similarity we are talking about is related to the pattern of conformal symmetry restoration and that in other aspects the high- T and high- μ systems are indeed very different.

Referee: In the Abstract, the authors write "This highly compressed matter may undergo a phase transition where neutron-rich nuclear matter melts into deconfined quark matter, liberating its constituent quarks and gluons [1]", where [1] is Shuryak's 1980 Phys. Rept. paper.

The authors are likely aware that the notion of the possible existence of quark matter in the cores of neutron stars was not initially proposed by Shuryak. Prior to his work, this idea had been put forth by:

Ivanenko, Dmitri D., and Kurdgelaidze, D. F. (1965) "Hypothesis concerning quark stars", *Astrophysics* 1 (4), 251–252

Ivanenko, Dmitri D.; Kurdgelaidze, D. F. (1969). "Remarks on quark stars", *Lettere al Nuovo Cimento* 2, 13–16

H. Fritzsche, M. Gell-Mann, and H. Leutwyler, *Phys. Lett.* 47B (1973) 365.

Our reply: We thank the referee for bringing to our attention these important works that we were unaware of. We have now included them as references 1-3.

Referee: Is there a specific motive behind the authors' choice to display the pressure as a function of the baryon chemical potential in Figure 4, instead of baryon number or energy density? By presenting pressure as a function of either baryon number or energy density, it would establish a more direct connection for many readers to the densities observed in the cores of neutron stars. This choice would likely be more illustrative and enhance the readers' comprehension.

Our reply: This was indeed a choice we carefully considered. The reason behind using the baryon chemical potential instead of the number density is to better facilitate a connection to the high-temperature case, where the thermodynamics of the quark-gluon plasma is always presented in terms of the intensive parameter T . It is of course true that the energy and number densities are more commonly used variables in the NS-physics context, but we felt that using those quantities in our other plots and indicating the central densities of NSs of various masses in fig. 4 should suffice to make the connection between the thermodynamical quantities and the number and energy densities sufficiently clear.

Referee: In Figures 5-7, it would be beneficial to indicate the central densities achieved in at least some of the M-R models in the corresponding pressure versus energy density plots.

Our reply: This is in principle a very useful suggestion. We have now indicated the central densities of TOV stars in our previous fig. 7 (now fig. 8), where this addition was both easily implementable and maximally interesting because the effect of the pQCD information is visible here.

Reply to referee 3:

We thank referee 3 for their valuable critical remarks. As we will detail below, we believe that the bulk of the criticism is, however, related to (very interesting and nontrivial) questions that have been successfully addressed in existing literature. Indeed, below we will make numerous references to four published articles, which we abbreviate as follows:

[KK] O. Komoltsev and A. Kurkela, Phys.Rev.Lett. 128 (2022) 20, 202701, arXiv:2111.05350 [nucl-th]

[GKK] T. Gorda, O. Komoltsev and A. Kurkela, Astrophys.J. 950 (2023) 2, 107, arXiv:2204.11877 [nucl-th]

[ER] L. Rezzolla and C. Ecker, Mon.Not.Roy.Astron.Soc. 519 (2022) 2, 2615-2622, arXiv: 2209.08101 [astro-ph.HE]

[GKKM] T. Gorda, O. Komoltsev and A. Kurkela and A. Mazeliauskas, JHEP 06 (2023) 002, arXiv: 2303.02175 [hep-ph]

Referee: The authors employ the assumption that at extremely high densities, more than 7 times the central density of any neutron star, the equation of state (EOS) can be predicted from perturbative-QCD (pQCD) calculations. Furthermore, the authors claim that this constraint on the equation of state constrains the equation of state within neutron stars. They have made this claim in the literature previously, for example Annala et al., Phys. Rev. Lett. 120, 172703 (2018); arXiv:1904.01354 (2019); Nat. Phys. 16, 907 (2020); Phys. Rev. X 12, 011058 (2022). If true, this represents an important consideration for modeling neutron stars. This paper is an extension of previous work by the group, in which the properties of pQCD matter is evaluated and methods of interpolating through the large density range from pQCD validity to neutron star densities are considered.

The authors claim their analysis is able to show that matter at the highest densities in neutron stars is consistent with that of quark matter. They further state their analysis demonstrates that quark matter most likely exists at the highest densities found in the most massive neutron stars. The latter is based on a statistical sampling of model equations of state (both parameterized and so-called non-parametric equations of state) constrained at low densities by chiral effective field theory calculations of neutron matter and at higher densities by observed properties of neutron stars.

As far as I can determine, the authors' results follow from their various assumptions, but I think that more consideration of the effects of these assumptions is warranted.

Our reply: We mostly agree with the referee's summary of our work, but disagree on a few important points that we will return to in more detail below but already briefly respond to here.

First, it should be noted that there are no specific nontrivial "assumptions" related to the claim that at densities of order $40n_s$ or higher, the EoS of dense QCD matter can be obtained from perturbative thermal quantum field theory.² At these densities, QCD is sufficiently weakly coupled so that the system is safely in the deconfined quark matter phase and that its thermodynamic properties can be approached with perturbative means.³ Whether or not these results are sufficiently accurate to have

²The only potentially nontrivial issue here is related to possible contributions from nonperturbative phenomena such as quark pairing, but as we discuss in detail e.g. in 0912.1856, such contributions are highly suppressed at baryon densities $n \gtrsim 40n_s$.

³A recent study of the convergence of the perturbative results using machine-learning tools can be found from [GKKM].

phenomenological impact on neutron-star physics is a separate question, but the pQCD constraint we employ stands on a solid ground.

Second, the fact that the pQCD limit quantifiably constrains the NS-matter EoS has been amply demonstrated in recent literature, although not in the papers the referee mentions but rather in the works [KK,GKK,ER] listed above. Below, we will summarize the findings of these papers in more detail, but the main conclusion is simple to state: the pQCD constraint plays an important role in the softening of the NS-matter EoS between the central densities of typical $1.4M_{\odot}$ pulsars and maximally massive stars.

Finally, and perhaps most importantly, it is crucial to note that the question studied in our present manuscript — is near-conformal QM present in the cores of massive NSs or not — is fully independent of the role the pQCD constraint plays in the inference of the NS-matter EoS. Indeed, should a combination of other constraints (say, the low-density nuclear-matter EoS and astrophysical data) be enough to demonstrate that NS matter behaves in a near-conformal fashion in the cores of maximally massive NSs, the result would be just as strong as the present one. The case for QM cores only relies on the inferred properties of matter at these densities, not on the particular theoretical or observational inputs that have contributed to this result.

Referee: The authors state that with few exceptions, model-agnostic studies of the neutron star matter EOS fail to take into account information from pQCD calculations. I don't think this omission is as serious as the authors suggest. All of these studies employ the constraint that matter remain causal at all densities. The squared speed of sound c_s^2 can be shown to approximate or exceed $1/3$ at densities around $3n_s$ (nuclear saturation density), which is about half the central density of the most massive stars, otherwise the EOSs won't be able to support 2 solar masses or more. At this approximate central density, stars typically have masses slightly larger than 1.4 solar masses (see Figure 1, for example). The properties of neutron stars with larger central densities and masses are relatively insensitive to the EOS at higher densities, as long as it remains causal and as long as 2 solar mass stars are able to be supported.

Certainly, the properties of 1.4 solar mass neutron stars, the most likely value for X-ray sources and merger components (which provide most of the available astrophysical information concerning radii), are completely determined by the EOS below this density. But studies have shown that even the properties of 2 solar mass stars are little influenced by the EOS above this density (unless a strong first-order phase transition appears just above this density). Therefore, whether or not the squared sound-speed limit above $40n_s$ is taken to be 1 or $1/3$ is not likely to be important. And at high densities, if c_s^2 is approximately unity, by implication γ becomes approximately 2, only slightly above the pQCD value.

Our reply: Once again, we want to emphasize that the question whether or not the pQCD limit meaningfully constrains the inferred properties of NS matter is not a central topic for the present article. However, as this question is nevertheless closely related to our work, in the following we will provide a brief summary of what is presently known about it.

We agree with the referee that knowledge of the speed of sound squared satisfying $c_s^2 \approx 1/3$ at $40n_s$ would alone not appreciably influence the EoS at neutron-star densities. This is, however, not

the essence of the pQCD constraint, which instead amounts to the joint requirement that the EoS is causal, thermodynamically consistent, and compatible with the pQCD limit *at all densities*. The derivation of this result was precisely the topic of the 2022 PRL [KK].

In [KK], the authors demonstrated in a rigorous and model-independent fashion that even when only applied from approx. 40 saturation densities onwards, the pQCD limit constrains the NS-matter EoS all the way down to $2.3n_s$. This perhaps somewhat counter-intuitive result arises not from the speed of sound, but from the requirement that the triplet (p, n_B, μ_B) can be causally connected to the high-density limit from any lower density. The admittedly technical but fully general construction leading to this conclusion can be found from [KK], while the quantitative impact of the pQCD constraint on the NS-matter EoS and its interplay with astrophysical constraints are further discussed in [GKK,ER].

A very important result that is demonstrated, e.g., in fig. 1 of [GKK] is that the pQCD constraint quantifiably softens the overall behavior of the NS-matter EoS across a wide density range, beginning well below the central densities of TOV stars. The exact same effect is visible in our fig. 7, where the impact of the pQCD constraint on various physical quantities is clearly seen in the last panel. In particular, we observe a considerable tightening of the posterior distribution of d_c for densities $n > 4n_s$, implying that the pQCD directly affects the likelihood of QM cores in TOV stars. While this panel corresponds to the GP method, the same effect is naturally present in our other results as well, but as the pQCD limit is by construction always present in the interpolated EoSs, these results are not available without the pQCD constraint.

In summary, the effect of the pQCD constraint on the NS-matter EoS has been robustly demonstrated in a series of earlier works [KK,GKK,ER]. The mechanism behind this perhaps counterintuitive result is not causality alone, but additionally relies on thermodynamic consistency [KK].

Referee: The authors need to make the case that imposition of the pQCD constraint actually does impact things like the radii of normal (1.3-1.6 solar mass) neutron stars as well as the maximum neutron star mass before claiming that “most likely” quark matter exists in the most massive neutron stars. One way to show this would be to make a test calculation in which pQCD was replaced by a sound speed of unity and, by implication, $\gamma = 2$, (at the same high densities equal to and exceeding $40n_s$), and employing the same astrophysical constraints. If the predictions of radii (or, equivalently, tidal deformabilities or moments of inertia) are thereby significantly affected, the authors’ case will have been made.

Our reply: Here, we return to topics partially covered above, but since they are presented in such a direct way in this comment, we address them in somewhat more detail here. We separate our answer to two parts, corresponding to the two claims made by the referee: 1) that we should demonstrate the impact of the pQCD constraint on the properties of NSs and the matter they contain, and 2) that the presence of quark matter inside massive NSs requires that the pQCD limit clearly affects the inference of NS (and NS matter) properties.

1. As mentioned above, this interesting question was precisely the topic of a number of earlier articles, including [KK,GKK,ER]. As can be seen from the results of these works as well as from our fig. 7, the pQCD constraint has a noticeable effect on the properties of matter in NS cores

Figure 2: A version of our fig. 1, where we have now included two new $1\text{-}\sigma$ level results using the GP framework: one with no pQCD limit implemented (black solid curves) and one with no astrophysical constraints except for the two-solar-mass limit (blue dashed curves). In the latter case, the $2M_{\odot}$ limit was implemented as a hard cut.

down to densities of a few n_s , although the effect is, of course, more pronounced in TOV stars. For the mass-radius relation, the effect is more suppressed than for the EoS, being mostly visible in the maximal mass of stable NSs.

For the convenience of the referee, we have now redone our GP analysis for the conformal distance d_c without the pQCD input. The black solid lines in the above fig. 2 correspond to this result, featuring a dramatic increase of the $1\text{-}\sigma$ upper limit for the quantity across a wide density range. It is thus undeniable that the pQCD constraint plays a major role in our setup, although we were only able to (easily) verify this for the GP computation.

2. While the answer to question 1 above was positive, we want to briefly return to our earlier argument concerning why the presence of QM inside NS cores in no way depends on the role that the pQCD constraint plays in EoS inference. An interesting reference perhaps further illuminating this point is provided by the Nature Physics article 1908.09728, where a closely analogous argument was made for the production of hot quark-gluon plasma in relativistic heavy-ion collisions. In that case, no input from high-temperature pQCD calculations was ever used, and the analysis instead entirely relied on numerical lattice field theory simulations and observational data. Similarly, should the low-density CEFT EoS and/or astrophysical data one day become accurate enough so that the high-density pQCD constraint would no longer give additional constraints on the properties of dense QCD matter at NS densities, this would in no way hinder drawing

positive conclusions on the presence of deconfined matter in massive NSs. This conclusion only relies on the inferred properties of NS matter displaying properties consistent with QM — not on the particular ingredients that led to this result.

Referee: Actually, a telling calculation would be to assume somewhat the opposite: remove all astrophysical constraints with the exception of a 2 solar mass lower limit to the maximum mass, but keep the low-density chiral EFT, causality, and the high-density pQCD constraints. What would the new preferred EOS and M-R bands compare to the ones in which other astrophysical constraints were kept? If the bands are very similar, this would seriously weaken the authors' conclusions.

Our reply: This question has been addressed for the EoS in fig. 4 of [GKK], utilizing the GP framework. For the convenience of the referee, we have now similarly computed the d_c parameter, shown as the blue dash-dotted line in fig. 2 above: in this result, all astrophysical observations except the 2 solar-mass one are removed, while the CEFT and pQCD constraints as well as causality are kept.

From these results, it is easy to conclude that it is precisely the interplay of all the different constraints, including both the pQCD limit and astrophysical measurements, that leads to our highly constrained final results, although for the M - R band the result is much less pronounced (see fig. 3 and our answers to the following three points below).

Referee: Another problem with the manuscript is that the authors do not show a mass-radius diagram, which is a standard representation for neutron stars. How does their preferred most-likely band compare to what is obtained by assuming the ultimate sound speed is unity instead of $1/3$?

Our reply: The mass-radius diagram is displayed in our figs. 5-7 using various interpolation methods and observational constraints; in particular, the modest effect of the pQCD constraint on the MR relation can be seen from the right-most column of fig. 7. Note again that the pQCD constraint does not amount to setting the speed of sound to any particular value but rather to demanding that the EoS can be connected to the pQCD limit in a causal and thermodynamically consistent fashion.

Referee: The role of astrophysical constraints is a little ambiguous. X-ray observations, in particular, are subject to large systematic uncertainties, especially those from quiescent and bursting sources where the distance uncertainties are important. If each observational constraint (with uncertainties) were shown in an M-R diagram, the reader could assess the significance of each constraint. One would have to include distance uncertainties in both quiescent and bursting sources, unlike what is often shown in the cited papers, which frequently assume a distance. Ideally, the significance of each type of observational constraint (pulsar mass measurements, tidal deformabilities, maximum mass upper limit from GW170817, NICER, bursting sources, and quiescent sources) would be delineated.

Our reply: The significance of the different inputs is examined in our figs. 6 and 7, while the individual X-ray measurements are displayed in fig. 8 of our earlier article 1903.09121. The importance and consistency of various measurements are furthermore thoroughly discussed in the Supplemental Material of 2008.12817.

We also note that all types of observational data we use include uncertainties which are marginalized

over in the Bayesian framework. Importantly, this includes the distance uncertainty of the X-ray measurements the referee mentions. Lastly, we note that we combine multiple measurement types (e.g., measurements of apparent areas from qLMBXs, cooling tail measurements from X-ray bursters, etc.) precisely to mitigate any systematic bias originating from one type of measurement method alone.

Referee: How important is each observational constraint to localizing the most-likely band in either the M-R diagram or the pressure-energy density diagram? And how does each observational constraint affect the likelihood of quark matter in the densest stars?

Our reply: As noted above, this question has already been partially addressed in our figs. 6 and 7. However, as this is admittedly a very interesting point, we have now performed an additional analysis (fig. 3 below) that displays our GP results obtained using the pQCD constraint together with various combinations of X-ray or GW measurements, dividing the X-ray measurements to two groups: NICER and non-NICER (see table II of the manuscript). The four columns in this figure stand for: everything (leftmost column), only non-NICER, only NICER, and NICER + GW170817 (including both tidal deformabilities and the likely collapse of the remnant into a black hole).

As we can see from this figure, the non-NICER X-ray measurements play a particularly important role in our final results, which is something we also highlight in the manuscript.

Referee: In conclusion, the authors need to back up their claim that including the pQCD constraint is essential to assessing the probability of a quark-hadron transition in the densest neutron stars. Replacing the pQCD constraint with another arbitrary constraint (such as a high-density sound speed limit of 1 or 2/3) should suffice. The role of astrophysical constraints (apart from that of imposing a minimum maximum mass) should also be clarified; their importance might be assessed from a calculation in which they are removed. Once these tests have been completed, this paper could be a valuable contribution to the study of quark matter in neutron stars. None of their previous papers have attempted such comparisons.

Our reply: The referee raised many important and nontrivial questions in their report, but as we have argued above, we believe that most of them have already been answered in existing literature as well as the Methods section of the current manuscript. In particular, the effect of the pQCD constraint on the inferred properties of NS-matter was demonstrated to be sizable in [KK,GKK,ER], but as we argue above, even this is not crucial for our present argument concerning the presence of deconfined matter in massive NSs. Finally, we note that the role of different astrophysical constraints on our results was already addressed in figures 6 and 7 of the earlier manuscript (7 and 8 in the updated version), and we have now further extended this analysis both above and in the updated draft.

While we strongly believe that a more detailed analysis of the effects of the pQCD constraint is not a good fit to the present manuscript, we agree with the referee that this topic could have been discussed (and referenced) more thoroughly and have now extended our discussion of this topic on p. 4. We hope that together with our detailed responses presented in this document, this change will convince the referee of the soundness of our work and of the quality of its presentation.

Figure 3: An extension of our original fig. 7, further dissecting the impacts of different types of X-ray measurements on our results. In response to one of the comments of referee 2, the central densities of TOV stars are indicated with the light grey columns here.

REVIEWER COMMENTS

Reviewer #1 (Remarks to the Author):

I thank the authors for carefully addressing my comments and providing extra information to concretely presenting their arguments. They have considered a large set of hadronic models available in CompOSE and compared the behavior of these models with their analysis. They have concluded that indeed some of the models could present near conformal symmetry but with a very negligible likelihood. They also show that the normalized pressure of hadronic models show a similar behavior to their result but conclude that the hadronic models "is already in its way down at the TOV density". They have clarified the validity of the cs4 with respect to the cs5 description. They have also clarified the points raised by the other referees. I would just like to leave these two comments:

- it is very nice that the authors have calculated the likelihood of the NM models considering their constraints. Note that CompOSE accepts all models and it is up to the user to choose the models considered more appropriate. The authors have chosen to include all of them with a likelihood according to their constraints. Many of them seem to fail at low densities probably due to the fact that they have not been fitted to chEFT PNM. These certainly gives them a small likelihood, although possibly not due to the high density behavior. In fact, the low density behavior the authors build is attached with the uncertainty connected with their ansatz to describe beta-equilibrium matter, besides the one that comes directly from the chEFT and the band obtained is not defined with a credible interval.

- It is interesting to see the similar behavior the hadronic models show for the normalized pressure with respect to the present results. Concerning the comment "is already on its way down at the TOV density", the GP description also shows a similar behavior, and even the cs4 calculation (before the rise to the pQCD values, in particular, it is very clear on the bottom limits of the distribution) . As the authors say, this may just indicate that matter is still strongly interacting. I consider that the authors argument concerning the flattening of the normalized pressure occurring above $\mu_B \sim 1250$ MeV is not so strong in view of the behavior of the hadronic models, since this flattening occurs in all the hadronic models and at densities where these models are still valid. I propose the authors also include the plot with the normalized pressure and the hadronic models in section F of the Methods, to show this behavior.

A consider the present work will bring a new insight into the possible occurrence of quark matter inside neutron stars and, therefore, I recommend its publication in Nature Communications.

Reviewer #2 (Remarks to the Author):

[Editorial Note: This reviewer provided remarks only to the Editor]

Reviewer #3 (Remarks to the Author):

The authors gave a quite thorough response to my comments, and alerted me to several papers that addressed some of the more important ones. With apologies for the extra time it took, I examined these papers in some detail in order to demonstrate to myself that my comments were, in fact, addressed. I am still not completely convinced that thermodynamic consistency implies constraints at extremely high density play a strong role at normal neutron star densities, but that does appear to be the case. With the additional caveats that now exist in the paper (some in response to the other referee comments), I think that the authors' conclusions are warranted. Furthermore, by performing simulations in which the various kinds of observations are selectively included, the authors addressed the other major concern I had. One is now able to appreciate with Figs. 7 and 8 the extent to which specific observations are influencing the results.

Thus, my most serious objections have been considered, and although I still have some misgivings, I think this paper represents a valuable addition to the literature and should be published.

Reply to reviewer 1:

We thank reviewer 1 for their insightful comments and for drawing our attention to the behavior of the normalized pressure that we indeed had failed to properly discuss. Below, we reply to all comments made and summarize the changes we have implemented in the updated draft.

Referee: I thank the authors for carefully addressing my comments and providing extra information to concretely presenting their arguments. They have considered a large set of hadronic models available in CompOSE and compared the behavior of these models with their analysis. They have concluded that indeed some of the models could present near conformal symmetry but with a very negligible likelihood. They also show that the normalized pressure of hadronic models show a similar behavior to their result but conclude that the hadronic models “is already in its way down at the TOV density”. They have clarified the validity of the cs4 with respect to the cs5 description. They have also clarified the points raised by the other referees. I would just like to leave these two comments:

- It is very nice that the authors have calculated the likelihood of the NM models considering their constraints. Note that CompOSE accepts all models and it is up to the user to choose the models considered more appropriate. The authors have chosen to include all of them with a likelihood according to their constraints. Many of them seem to fail at low densities probably due to the fact that they have not been fitted to chEFT PNM. These certainly gives them a small likelihood, although possibly not due to the high density behavior. In fact, the low density behavior the authors build is attached with the uncertainty connected with their ansatz to describe beta-equilibrium matter, besides the one that comes directly from the chEFT and the band obtained is not defined with a credible interval.
- It is interesting to see the similar behavior the hadronic models show for the normalized pressure with respect to the present results. Concerning the comment “is already on its way down at the TOV density”, the GP description also shows a similar behavior, and even the cs4 calculation (before the rise to the pQCD values, in particular, it is very clear on the bottom limits of the distribution) . As the authors say, this may just indicate that matter is still strongly interacting. I consider that the authors argument concerning the flattening of the normalized pressure occurring above $\mu_B \sim 1250$ MeV is not so strong in view of the behavior of the hadronic models, since this flattening occurs in all the hadronic models and at densities where these models are still valid. I propose the authors also include the plot with the normalized pressure and the hadronic models in section F of the Methods, to show this behavior.

I consider the present work will bring a new insight into the possible occurrence of quark matter inside neutron stars and, therefore, I recommend its publication in Nature Communications.

Our reply: In their first bullet point above, the referee speculates that a differing low-density behavior might explain the small likelihoods that are assigned to certain nuclear-matter model EoSs in our analysis. We can, however, confirm that this is not the case: the likelihoods are assigned only based on tests against astrophysical data and whether the high-density limit of the EoS can be connected to the pQCD result in a thermodynamically consistent fashion, i.e. agreement with our low-density EoS

(from CEFT) is not tested at all. We have now changed a possibly misleading wording in the caption of Fig. 5 and added a few explanatory words to the corresponding discussion in the text.

As to the latter comment, we agree with the referee that adding the pressure figure to the Methods section adds value to the paper, and have now done so. In our opinion, there exists a qualitative difference in the overall trends of p/p_{free} between the nuclear-matter models and our results: for the vast majority of the model EoSs, this quantity decreases at the TOV point, while only a small fraction of the interpolated EoSs feature similar behavior at any density. Having said this, we have now softened our choice of words in footnote 2 of the main text and added a few carefully formulated sentences to the Methods section describing the new Fig. 6. We feel that this allows the readers to draw their own conclusions from the figure.

Reply to reviewer 3:

Referee: The authors gave a quite thorough response to my comments, and alerted me to several papers that addressed some of the more important ones. With apologies for the extra time it took, I examined these papers in some detail in order to demonstrate to myself that my comments were, in fact, addressed. I am still not completely convinced that thermodynamic consistency implies constraints at extremely high density play a strong role at normal neutron star densities, but that does appear to be the case. With the additional caveats that now exist in the paper (some in response to the other referee comments), I think that the authors' conclusions are warranted. Furthermore, by performing simulations in which the various kinds of observations are selectively included, the authors addressed the other major concern I had. One is now able to appreciate with Figs. 7 and 8 the extent to which specific observations are influencing the results.

Thus, my most serious objections have been considered, and although I still have some misgivings, I think this paper represents a valuable addition to the literature and should be published.

Our reply: We would like to sincerely thank reviewer 3 for carefully studying the references we mentioned in our previous reply and for presenting this favorable and encouraging evaluation of our work.

REVIEWERS' COMMENTS

Reviewer #1 (Remarks to the Author):

The authors have clarified both comments and completed the discussion accordingly. In particular, they now include in the methods a new figure (fig. 6) showing the behavior of p/p_{free} for a large set of hadronic EoS, and clarify how the likelihoods were assigned to these EoS.

As already stated in the last report, I consider the present work will bring a new insight into the possible occurrence of quark matter inside neutron stars and, therefore, I recommend its publication in Nature Communications.

Reply to reviewer 1:

Referee: The authors have clarified both comments and completed the discussion accordingly. In particular, they now include in the methods a new figure (fig. 6) showing the behavior of p/p_{free} for a large set of hadronic EoS, and clarify how the likelihoods were assigned to these EoS. As already stated in the last report, I consider the present work will bring a new insight into the possible occurrence of quark matter inside neutron stars and, therefore, I recommend its publication in Nature Communications.

Our reply: We thank reviewer 1 for their kind evaluation of our work and their recommendation for its publication.